# Prion protein promotes copper toxicity in Wilson disease

Raffaella Petruzzelli [1,2,13], Federico Catalano [1,3,13], Roberta Crispino[1], Elena V. Polishchuk[1], Mariantonietta Elia[1], Antonio Masone [4], Giada Lavigna[4], Anna Grasso [4], Maria Battipaglia[1], Lucia Vittoria Sepe[1], Banu Akdogan[5], Quirin Reinold[6], Eugenio Del Prete [1], Diego Carrella[1], Annalaura Torella[1,7], Vincenzo Nigro[1,7], Enrico Caruso [8], Nicole Innocenti [9], Emiliano Biasini [9], Ludmila V. Puchkova [10,11], Alessia Indrieri [1,12], Ekaterina Y. Ilyechova[10,11], Pasquale Piccolo [1], Hans Zischka[5,6], Roberto Chiesa [4] ✉ & Roman S. Polishchuk [1] ✉

Copper (Cu) is a vitally important micronutrient, whose balance between essential and toxic levels requires a tightly regulated network of proteins. Dysfunction in key components of this network leads to the disruption of Cu homeostasis, resulting in fatal disorders such as Wilson disease, which is caused by mutations in the hepatic Cu efflux transporter ATP7B. Unfortunately, the molecular targets for normalizing Cu homeostasis in Wilson disease remain poorly understood. Here, using genome-wide screening, we identified the cellular prion protein (PrP) as an important mediator of Cu toxicity in WD. Loss of ATP7B stimulates hepatic expression of PrP, which promotes endocytic Cu uptake, leading to toxic Cu overload. Suppression of PrP significantly reduces Cu toxicity in cell and animal models of Wilson disease. These findings highlight the critical regulatory role of PrP in copper metabolism and open new avenues for exploring the therapeutic potential of PrP suppression in Wilson disease.

Copper (Cu) is an indispensable micronutrient for the growth and replication of all eukaryotic organisms[1]. By exploiting the ability of copper to switch between $Cu^+$ and $Cu^{2+}$ oxidation states, cuproenzymes drive vitally important metabolic processes including respiration, anti-oxidant defense, biosynthesis of neuropeptides, and components of connective tissue[1,2]. However, excess copper represents a serious danger for cells and organisms due to the ability of Cu

to provoke extensive damage via activation of specific cell death mechanisms[3–5]. Thus, cells and organisms have evolved a finely tuned network of transporters and copper-binding proteins to provide a sufficient supply of the metal and to avoid toxicity[1,2,6].

A failure of this regulatory network to properly support Cu homeostasis manifests in serious life-threatening disorders. Mutations in transporters, which drive adsorption of dietary Cu, cause severe

[1]Telethon Institute of Genetics and Medicine (TIGEM), Pozzuoli, Italy. [2]Genomics and Experimental Medicine Program, Scuola Superiore Meridionale (SSM, School of Advanced Studies), Naples, Italy. [3]Institute of Biosciences and Bioresources, National Research Council, Naples, Italy. [4]Department of Neuroscience, Laboratory of Prion Neurobiology, Istituto di Ricerche Farmacologiche Mario Negri IRCCS, Milan, Italy. [5]Institute of Molecular Toxicology and Pharmacology, Helmholtz Center Munich, German Research Center for Environmental Health, Neuherberg, Germany. [6]Institute of Toxicology and Environmental Hygiene, Technical University Munich School of Medicine and Health, Munich, Germany. [7]Department of Precision Medicine, University of Campania "Luigi Vanvitelli", Naples, Italy. [8]Department of Biotechnology and Life Sciences, University of Insubria, Varese, Italy. [9]Department of Cellular, Computational and Integrative Biology, University of Trento, Povo, TN, Italy. [10]Department of Molecular Genetics, Research Institute of Experimental Medicine, St. Petersburg, Russia. [11]ITMO University, St. Petersburg, Russia. [12]Institute for Genetic and Biomedical Research (IRGB), National Research Council (CNR), Milan, Italy. [13]These authors contributed equally: Raffaella Petruzzelli, Federico Catalano. ✉e-mail: roberto.chiesa@marionegri.it; polish@tigem.it

systemic Cu deficiencies[7,8]. Altered expression of Cu transporters in several cancers has been also associated with elevated Cu supply, fueling different processes that support tumor growth[9]. Finally, Cu toxicity occurs in Wilson disease (WD), which is caused by mutations in the *ATP7B* gene encoding a hepatic Cu transporter that drives excretion of excess Cu to the bile[10]. Loss of *ATP7B* function leads to Cu accumulation initially in the liver and subsequently in the brain thereby causing serious hepatic and neurological abnormalities culminating in death[10]. Approved treatments for WD have serious drawbacks in a substantial cohort of patients[11] but the development of new therapeutic strategies is hampered by the poor understanding of the molecular mechanisms that can be exploited to normalize Cu homeostasis.

## Results

### Screening for genes that promote copper toxicity in a cell model of Wilson disease

To find new genes that might be targeted for reduction of Cu toxicity in Wilson disease we employed genome-wide shRNA screening in ATP7B-KO HepG2 cells, a bona fide WD cell model[12,13]. These cells do not express ATP7B (Supplementary Fig. 1a, b) and consequently exhibit elevated sensitivity to copper (Supplementary Fig. 1c–e). We reasoned that shRNAs which reduce the ability of Cu to kill ATP7B-KO cells could identify gene targets to counteract copper toxicity in WD. We screened a pooled lentiviral library targeting 18,205 human genes. The library contained 10 pools, with 4–7 shRNAs for each gene distributed across the different pools. ATP7B-KO cells were transduced with different shRNA library pools and either left untreated (control) or treated with 0.5 mM $CuCl_2$ for 24 h (Fig. 1a). Genomic DNA was then isolated from the cells and read-counts for each shRNA were obtained by deep sequencing to determine the abundance of each shRNA in the Cu-treated cells compared to controls (Fig. 1a). Subsequent analysis identified enriched shRNAs corresponding to genes whose downregulation promoted survival of ATP7B-KO cells upon Cu overload (Fig. 1b). We assumed that some shRNAs might favor survival of ATP7B-KO cells by suppressing generic cell death genes rather than genes that specifically regulate Cu toxicity. To test this, we ran the same screen with the proapoptotic drug vinblastine instead of $CuCl_2$. Genes whose suppression allowed cells to resist Cu but not vinblastine (Supplementary Dataset 1) were considered as Cu-specific hits and studied further, while genes corresponding to shRNAs conferring resistance to both vinblastine and copper were discarded (see vinblastine screening results in Supplementary Dataset 2). We expected that genes encoding known Cu importers would be among the hits because their silencing should promote tolerance of ATP7B-KO cells to Cu. Indeed, two shRNAs specific for the divalent metal transporter DMT1 (also known as SLC11A2) significantly improved resistance to Cu (Supplementary Dataset 1). This finding supported the validity of our screening approach. Surprisingly, another expected Cu transporter, CTR1 (SLC31A1), did not appear among the hits. However, we found that CTR1 expression was markedly downregulated in ATP7B-KO cells compared to parental HepG2 cells and its genetic suppression did not offer any further advantage for Cu detoxification (see below). Other CD44-mediated copper uptake mechanisms have been recently identified and their role in cancer demonstrated[14] but CD44 was not in our hit list apparently due to a low expression in ATP7B-KO HepG2 cells[13]. Finally, we paid particular attention to recently reported genes promoting cell death via Cu-specific cuproptosis pathway[5]. Of these, LIPT1 and a member of the pyruvate dehydrogenase family, PDHX, were present in the list of enriched shRNAs. However, none of the LIPT1-specific shRNAs was enriched at least 2-fold and so did not qualify as a hit, while PDHX qualified as a bona fide hit (Supplementary Dataset 1), again indicating the robustness of the screen.

Gene ontology (GO) analysis of the hits revealed an enrichment in genes associated with transition metal transport/homeostasis (Fig. 1c

and Supplementary Dataset 3). Considering that protein products of some of these transition metal homeostasis genes are capable of handling copper (Supplementary Dataset 3), we decided to focus on this group of genes, subjecting them to secondary screens. Each gene was silenced with specific siRNAs and the resistance of silenced ATP7B-KO cells to Cu was evaluated by the MTT viability assay[15] (Fig. 1d). Confirmed hits were further validated using live/dead staining combined with automated high-content imaging (Fig. 1e). Several transition metal-transporting genes including *SLC11A2* (DMT1), *TFR2*, *SLC25A37*, *STEAP1*, *MMGT1* and *PRNP* were identified as robust targets for reduction of Cu toxicity (Fig. 1e). The *PRNP* gene encoding the prion protein (PrP) and the *TFR2* gene encoding transferrin receptor 2 emerged as the strongest hits whose suppression was very effective in protecting ATP7B-KO cells from Cu (Fig. 1f).

### Suppression of PrP improves resistance of ATP7B-deficient cells to Cu

Among the hits validated in secondary assays, we focused on candidates that may be targeted for therapy. For each hit, several parameters were considered for further investigation such as the robustness of the corresponding siRNA in suppressing Cu toxicity in ATP7B-KO cells, its involvement in copper metabolism, the absence of severe phenotypes in KO mice, and whether any human disease is associated with a loss of gene function (Supplementary Table 1). The *PRNP* gene encoding prion protein (PrP) emerged as the most attractive candidate. PrP is a GPI-anchored cell surface glycoprotein that has received considerable attention due to its central role in the pathogenesis of prion disease[16]. No disease has been attributed to the loss of PrP expression in humans[16] and loss of one functional *PRNP* allele is compatible with life[17]. *Prnp* knockout mice do not manifest any severe phenotype, showing only sensorimotor defects at very old age, which however are not seen in heterozygotes[18,19]. This indicates that PrP suppression should be safe in terms of possible undesired side effects. Previous research clearly demonstrated a link between PrP and Wilson disease[20–22]. Cu has been shown to stimulate expression of PrP[23,24], which contains several Cu-binding sites and has been suggested to operate as an endocytic receptor for Cu[25–29]. Therefore, we reasoned that PrP might stimulate Cu uptake and, thereby, toxicity in ATP7B-deficient cells, while PrP suppression could counteract this process.

First, we further validated the role of PrP in dose-response experiments. *PRNP* was silenced in ATP7B-KO cells and the resistance to different concentrations of Cu was evaluated. Both live/dead and MTT assays indicated that *PRNP* silencing improved resistance of ATP7B-deficient cells to Cu (Fig. 2a–c). In parallel, a fluorescent sensor of caspase 3/7 activity showed that *PRNP* silencing significantly reduced the number of apoptotic cells upon exposure to Cu (Supplementary Fig. 2). Second, we evaluated Cu content in *PRNP*-silenced cells. PrP depletion in ATP7B-KO cells led to a decrease in Cu levels as shown using the fluorescent CF4 copper sensor and by ICP-OES (Fig. 2d–f). In addition, PrP suppression reduced Cu-mediated transactivation of *MT1X* (Fig. 2g), which has been recently proposed as a transcriptional reporter for Cu because its expression strongly correlates with intracellular Cu levels[30]. Finally, considering that mitochondria emerged as the most sensitive organelle to Cu toxicity in Wilson disease[13,31], we investigated the impact of PrP suppression on mitochondrial function. Labelling with the membrane potential dye TMRE revealed that *PRNP* silencing prevented Cu-induced mitochondrial dysfunction in ATP7B-KO cells (Fig. 2h, i). These observations were further supported by electron microscopy (EM), which showed reduced mitochondrial damage in *PRNP*-silenced ATP7B-KO cells treated with Cu (Fig. 2j, k).

The role of PrP in promoting copper (Cu) toxicity was further demonstrated in primary hepatocytes. To this end, the Atp7b knockout (*Atp7b⁻/⁻*) mouse strain, representing a bona fide model of WD[32], was crossed with *Prnp⁻/⁻* mice to suppress PrP expression. Primary

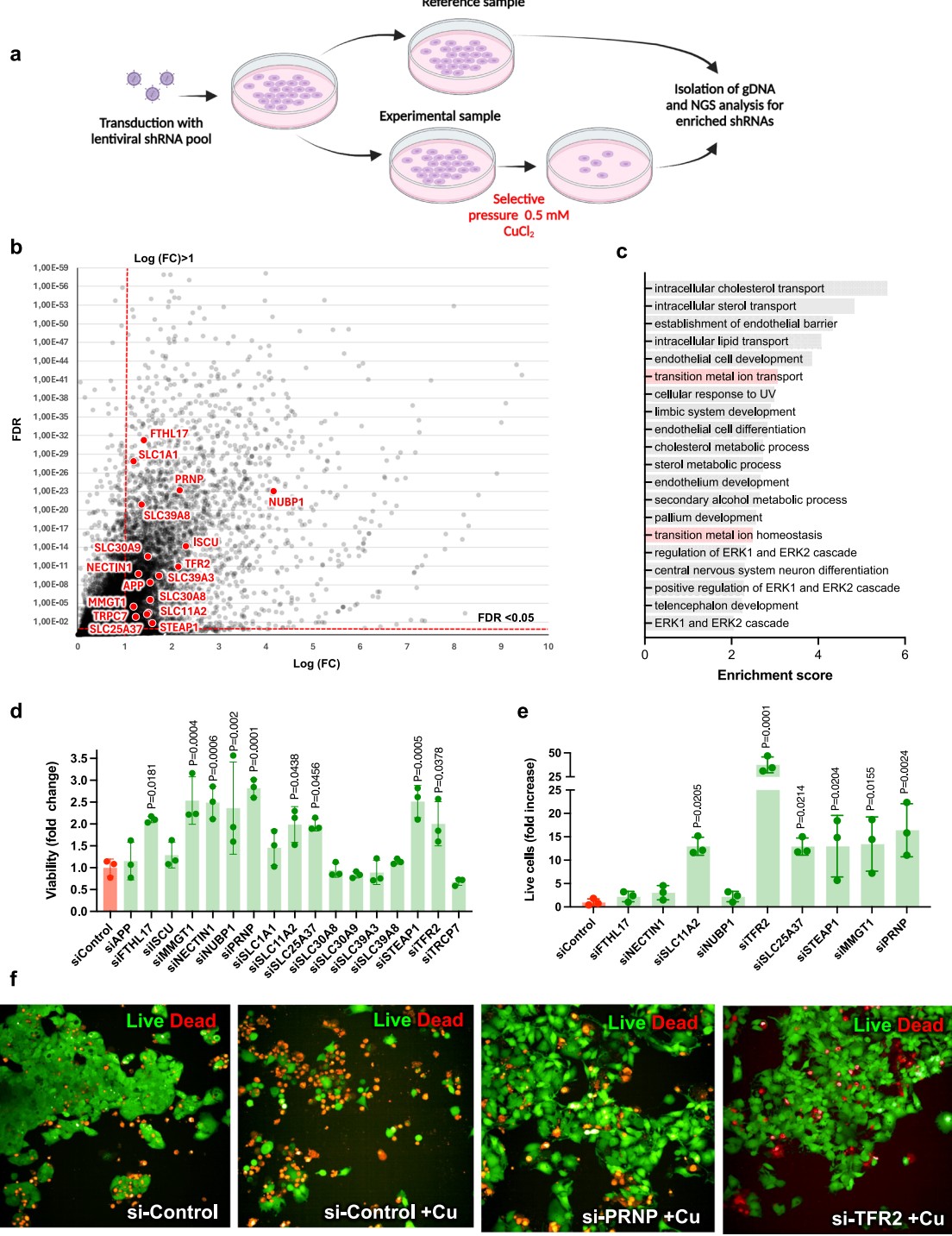

**Fig. 1 | Genome-wide shRNA screening and validation of hits reducing Cu toxicity in ATP7B-KO cells. a** Schematic representation of the screening workflow (Created in BioRender; https://BioRender.com/m58f469). **b** Each dot in the plot shows shRNAs enriched in the Cu-treated ATP7B-KO cells. *X*-axis – Log (base 2) of count fold change (FC) for each shRNA in Cu-treated cells compared to untreated. *Y*-axis – false discovery rate (FDR) value significance level for each shRNA. FDR levels below 0.05 and Log FC levels above 1 (red dash lines) were considered for hit selection. ShRNAs corresponding to genes involved in transition metal homeostasis (see **c**) are highlighted in red. **c** Screening hit genes (see supplementary dataset 1) were subjected to gene ontology (GO) enrichment analysis (see methods). The plot shows top 20 biological process terms that are significantly enriched among hit genes. GO categories related to transition metal homeostasis and transport are

highlighted in pink. **d** Transition metal hit genes (red text **b**) were silenced by specific siRNAs in ATP7B-KO cells for 48 h. Then the cells were incubated with 0.5 mM CuCl$_2$ for 24 h and their viability was evaluated by MTT assay (Mean ± SD; one-way ANOVA; *n* = 3 experiments). SiRNAs improving resistance to Cu were further validated using live-dead assay (see **e** and **f**). **e, f** The cells were treated as indicated in (**d**) then stained using live-dead fluorescent kit and analyzed using HCS Opera system (see methods). The graph in panel e shows % of live (green) cells for each treatment (Mean ± SD; one-way ANOVA; *n* = 3 experiments). Representative images in (**f**) demonstrate toxic impact of Cu resulting in increase of the dead (red) cells, while PRNP- and TFR2-specific siRNAs substantially improve resistance of the cells to Cu. Scale bar: 210 µm (**f**).

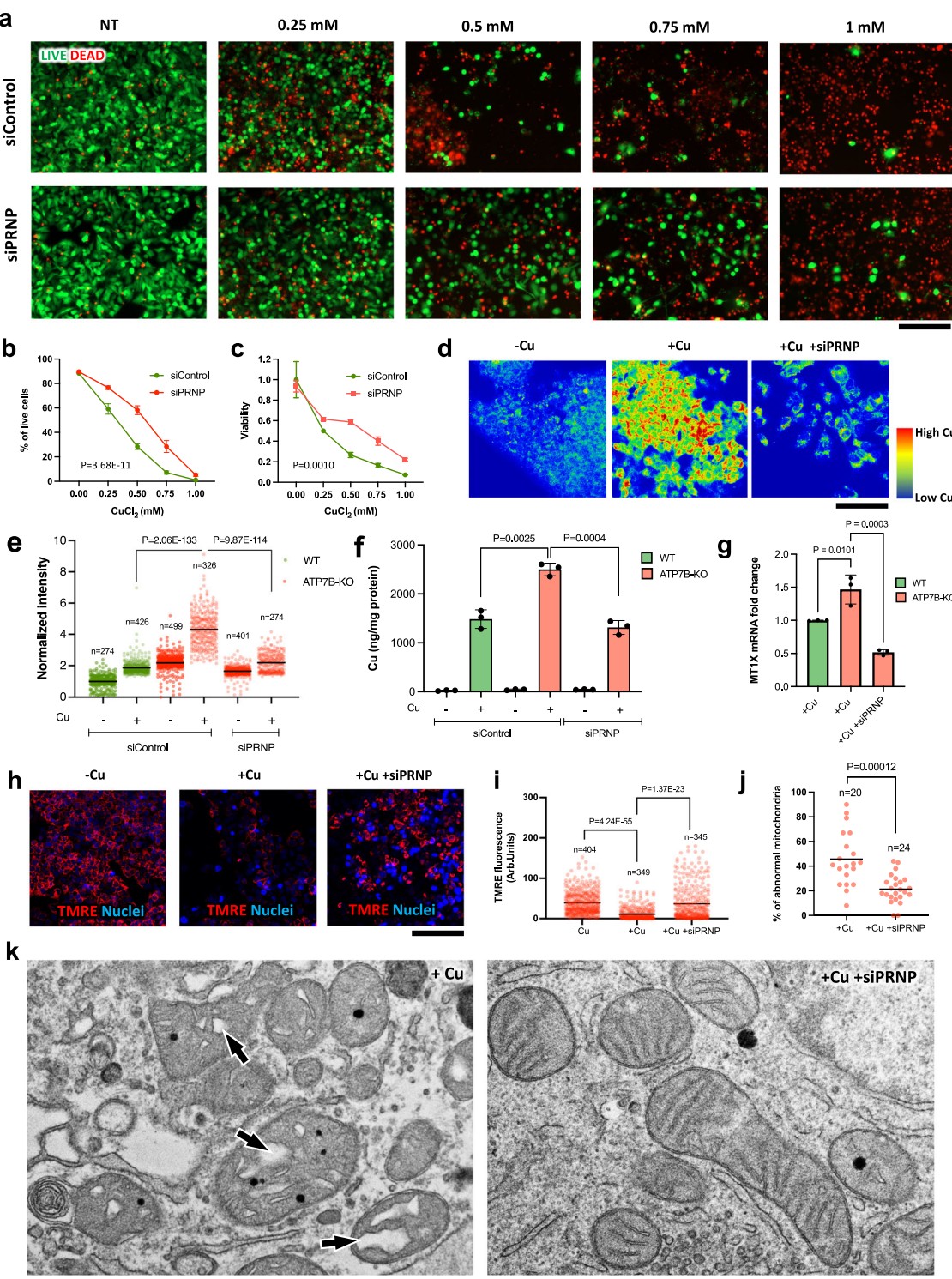

hepatocytes were then isolated from single knockout (*Atp7b*[−/−]) or double knockout (*Atp7b*[−/−] and *Prnp*[−/−]) mice and treated with Cu. Live/dead and apoptotic assays indicated that *Prnp* knockout reduced the mortality of *Atp7b*-deficient hepatocytes in response to Cu (Supplementary Fig. 3a, b). These observations were further supported by quantification, which indicated a decrease in the percentage of dead and apoptotic cells in *Atp7b*[−/−]:*Prnp*[−/−] hepatocyte populations compared to hepatocytes isolated from *Atp7b*[−/−] mice (Supplementary Fig. c, d).

In addition to genetic suppression, we tested whether the G54S and M128V polymorphic variants of PrP, that had been proposed to influence the clinical manifestations of WD[20,22,33] modulated Cu toxicity

in ATP7B-deficient cells. The corresponding mouse PrP variants (G53S or M128V) were transfected into ATP7B-KO HepG2 cells with stable *PRNP* knockdown. Cells were exposed to Cu and their viability was assessed after 24 h. We found that Cu treatment led to higher mortality in cells expressing the G53S variant of PrP compared to those expressing the WT protein (Supplementary Fig. 4), indicating that the G/S substitution in PrP promotes Cu toxicity. In contrast, expression of the M128V PrP variant did not significantly alter cell susceptibility to Cu toxicity (Supplementary Fig. 4).

Next, we asked whether pharmacological suppression of PrP improved tolerance to Cu in ATP7B-KO cells. To this end, we used small

**Fig. 2 | Silencing of *PRNP* reduces Cu toxicity in ATP7B-KO HepG2 cells.**
**a, b** ATP7B-KO cells were incubated with control (siControl) or *PRNP*-specific siR-NAs and exposed to CuCl₂ for 24 h. The cells were then assayed using a live/dead fluorescent kit. Representative micrograms (**a**) and their quantification (**b**) show that PRNP suppression increases % of live cells upon exposure to Cu (Mean ± SEM, two-way ANOVA; *n* = 5 view fields). **c** ATP7B-KO cells were treated as described in (**a, b**) and cell viability was evaluated using the MTT assay, showing that *PRNP*-silenced cells have higher viability upon exposure to Cu (Mean ± SEM; two-way ANOVA; *n* = 3 experiments). **d, e.** WT and ATP7B-KO cells were incubated with control or *PRNP*-specific siRNAs, exposed to 0.5 mM CuCl₂ (24 h), and stained with the fluorescent copper sensor CF4. Representative false color images (**d**) and quantification (**e**) show a reduction of the CF4 signal in *PRNP*-silenced cells upon Cu treatment (Mean; one-way ANOVA; n=cell number). **f** WT and KO cells were treated as indicated above (**d, e**) and then subjected to ICP-OES analysis. *PRNP* silencing reduces Cu levels in ATP7B-KO cells (Mean ± SD; one-way ANOVA; *n* = 3 experiments). **g** WT and KO cells were treated as indicated in (**d, e**) and then prepared for qRT-PCR to evaluate MT1X expression, which was significantly reduced by *PRNP*-specific siRNAs (Mean ± SD; one-way ANOVA; *n* = 3 experiments). ATP7B-KO cells incubated with control (siControl) or *PRNP*-specific siRNAs were exposed to 0.5 mM CuCl₂ for 24 h and then either labelled with the mitochondrial membrane potential dye TMRE (**h, i**) or prepared for electron microscopy (**j, k**). Representative micrograms (**h**) and quantification (**i**) show that *PRNP* suppression increases the TMRE signal in Cu-treated cells (Mean; one-way ANOVA; *n* = cell number) indicating improvement of mitochondrial function. Quantification (**j**) and representative EM images (**k**) indicate that *PRNP* silencing reduces number of abnormal mitochondria in Cu-treated ATP7B-KO cells (Mean; two-tailed *t*-test; *n* = cell number). Arrows in k (left panel) indicate swelling of mitochondrial cristae. Scale bars: 300 μm (**a, h**); 110 μm (**d**); 260 nm (**k**).

molecules that reduce PrP expression by different mechanisms. First, we used tetracationic zinc (Zn(II)-BnPyP) or copper (Cu(II)-BnPyP) porphyrins whose binding to PrP induces its rapid endocytosis and lysosomal degradation[34]. Pretreatment with Zn(II)-BnPyP or Cu(II)-BnPyP significantly reduced PrP levels and Cu toxicity in ATP7B-KO cells (Supplementary Fig. 5a–e), while the control Fe(III)-BnPyP, which does not downregulate PrP[34], had no effect on cell resistance to Cu. Further, we suppressed PrP expression with SM875, a compound that impairs folding of newly-synthesized PrP, thus inducing degradation of misfolded PrP molecules[35]. We found that SM875 reduced PrP levels and improved the resistance of ATP7B-KO cells to Cu (Supplementary Fig. 5f–h). Notably, the ability of chemical PrP degraders to counteract Cu toxicity in ATP7B-KO cells was paralleled by a decrease in bioavailable Cu levels as shown by the reduction in Cu-induced MT1X expression (Supplementary Fig. 5i).

## Loss of ATP7B leads to elevated PrP expression in hepatic cells

Although the Cu-dependent properties/functions of PrP were discovered more than two decades ago[16,26–28], its role in WD has been neglected due to its low expression in the liver[36,37]. However, Cu has been reported to stimulate *PRNP* transcription[23,24]. Therefore, we reasoned that Cu accumulation in ATP7B-deficient hepatic cells might elevate PrP expression. Indeed, we found that Cu induced *PRNP* in expression in ATP7B-KO cells (Fig. 3a), which also exhibited higher levels of PrP protein compared to parental WT HepG2 cells (Fig. 3b, c).

Next, we tested whether the loss of ATP7B affected PrP expression in vivo. To this end, PrP expression was assessed in the liver of Atp7b⁻/⁻ mice. Both qRT-PCR and Western blot showed higher PrP expression in the liver of Atp7b⁻/⁻ mice compared to control Atp7b⁺/⁻ animals (Fig. 3d–f). This was further confirmed by immunohistochemical labelling of PrP (Fig. 3g). Finally, we mined already published transcriptomics data in WD patients[38] and found higher *PRNP* mRNA levels compared to healthy liver tissue (Fig. 3h). Collectively, these data suggest that an increase in Cu upon ATP7B loss stimulates PrP expression, which in turn favors further Cu uptake/accumulation leading to higher PrP expression, thus generating a vicious circle (Fig. 3i).

How does accumulating Cu induce PrP expression in ATP7B-deficient cells? The *PRNP* promoter contains metal-responsive elements that are targeted by the metal-sensitive transcription factor MTF1, which transactivates *PRNP* in response to Cu[23]. To test whether Cu acts through MTF1 to stimulate *PRNP* expression in ATP7B-KO cells, we evaluated the impact of MTF1 silencing on Cu-dependent *PRNP* expression. We found that MTF1-specific siRNAs blocked the ability of Cu to induce *PRNP* expression (Fig. 3j). Thus, Cu stimulates MTF1-mediated transcription of *PRNP*, leading to higher PrP expression in ATP7B-KO cells.

The increase in PrP levels appears to allow excess Cu to bypass defense mechanisms activated by ATP7B-KO cells to limit Cu uptake,

such as the downregulation of the high-affinity copper transporter CTR1[13,39]. Supporting this, we found that CTR1 levels in ATP7B-KO cells were lower than in WT cells (Supplementary Fig. 6a). Moreover, further suppression of CTR1 did not alter the susceptibility of ATP7B-KO cells to Cu (Supplementary Fig. 6b), failing to provide any additional defense against Cu in ATP7B-KO cells. This may explain why shRNAs targeting CTR1 were not enriched in our initial screening for suppressors of Cu toxicity.

## PrP-mediated Cu toxicity requires Cu-binding sites

Our findings clearly indicate that PrP promotes Cu toxicity in ATP7B-KO cells. However, the mechanisms used by PrP to facilitate Cu toxicity remain unclear. To uncover these mechanisms, we first tested whether the Cu-binding sites of PrP are needed to promote Cu-mediated damage of ATP7B-KO cells. PrP has several histidine (H)-containing Cu-binding sites in its N-terminal octapeptide repeat (OR) region and in the more distal non-OR region[25,26,29]. To prevent Cu binding to these sites, we generated several PrP mutants in which all of the octapeptide repeats were deleted (PrP-ΔOR), all histidine residues in the OR region were replaced with alanines (PrP-Ala1), or the histidine residues in both the OR and non-OR regions were substituted with alanine (PrP-Ala2) (Fig. 4a). These PrP mutants were expressed in ATP7B-KO cells and their ability to stimulate Cu-mediated cell death was compared to that of the WT protein. We found that while overexpression of WT PrP significantly reduced tolerance of cells to Cu, all PrP constructs with mutated/deleted Cu-binding sites failed to do so (Fig. 4b, c), indicating that Cu binding to PrP is needed to promote Cu toxicity.

Next, we asked whether the promotion of Cu toxicity in ATP7B-KO cells by PrP was associated with endocytosis of the protein. Unfortunately, blocking PrP endocytosis in a specific manner is challenging because endocytic internalization of PrP occurs via different pathways and relies on different interactors[40–43]. Therefore, we took advantage of the ability of Cu to stimulate PrP endocytosis. Several studies indicate that Cu binding changes PrP conformation, favoring its endocytic internalization and, thereby, Cu uptake[27,29]. We asked how mutation/deletion of Cu-binding sites in PrP-ΔOR, PrP-Ala-1, and PrP-Ala-2 affected PrP endocytosis in ATP7B-KO cells. Remarkably, the failure of these PrP mutants to promote Cu toxicity was paralleled by their impaired endocytosis in response to Cu (Fig. 4d, e). In contrast, a significant amount of endocytosed WT PrP was detected in Cu-treated cells (Fig. 4d, e). This suggests that PrP endocytosis is needed to promote Cu toxicity because the inability of PrP mutants to be endocytosed correlates with their failure to stimulate Cu toxicity in ATP7B-KO cells.

Next, we reasoned that PrP distribution at the surface of polarized hepatocytes might be critical for its ability to drive Cu uptake in a physiological context. Given that Cu is supplied to hepatocytes from the blood, PrP must be situated on the sinusoidal (basolateral) surface of hepatocytes to facilitate the import of Cu from the bloodstream.

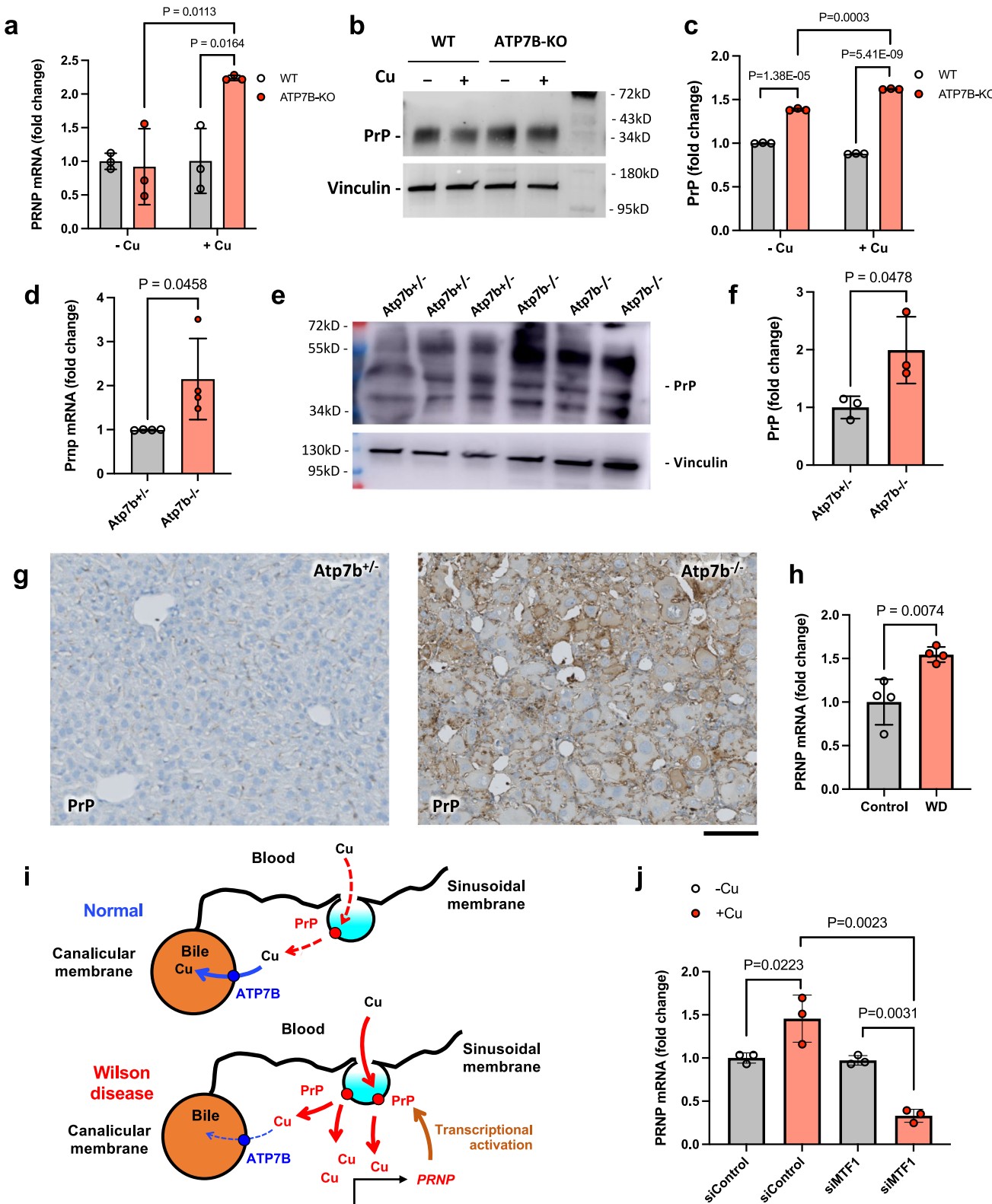

Basolateral localization of PrP has been documented in different polarized epithelial cells[42,44] but not in hepatic cells. To assess basolateral/apical localization of PrP, we polarized ATP7B-KO HepG2 cells with oncostatin M, which induces formation of apical vacuoles containing actin-rich microvilli reminiscent of biliary canaliculi in the liver (Fig. 4f). Confocal microscopy analysis showed that a significant pool of PrP resided outside these apical vacuoles in ATP7B-KO cells (Fig. 4f). Such basolateral localization should allow PrP to capture Cu from the

blood and transport it into hepatocytes. Indeed, we found that Cu promoted basolateral endocytosis of PrP in polarized hepatic cells (Fig. 4g). Further, the basolateral localization of PrP in hepatocytes was confirmed in the liver of Atp7b$^{-/-}$ mice by overlap with the basolateral marker Na/K-ATPase (Fig. 4h, i). In addition, PrP was detected in vesicular intracellular structures resembling endosomes (Fig. 4j), indicating that its basolateral endocytosis is likely to occur in hepatocytes of Atp7b$^{-/-}$ mice.

**Fig. 3 | ATP7B loss leads to elevated PrP expression in hepatic cells.** WT and ATP7B-KO HepG2 cells were treated with 0.5 mM CuCl$_2$ for 24 h and then prepared for qRT-PCR (**a**) or Western blot (**b, c**) to assess *PRNP*/PrP expression levels. Graph in (**a**) shows that Cu treatment strongly stimulates *PRNP* expression in ATP7B-KO cells but now in parental WT cell line (Mean ± SD; two-way ANOVA; *n* = 3 experiments). *Prnp*/PrP expression was analyzed in the liver of Atp7b$^{-/-}$ mice and control Atp7b$^{+/-}$ mice by qRT-PCR (**d**), Western blot (**e, f**), or immuno-histochemistry (**g**). All three methods demonstrate elevated expression of PrP in the liver of Atp7b$^{-/-}$ mice (Mean ± SD; *t*-test; *n* = 4 animals for **d**, *n* = 3 animals for **f**). **h** Expression levels of *PRNP* in the healthy or WD-affected human liver tissue were extracted from transcriptomics data published by ref. 38. The graph shows higher expression of *PRNP*

in the WD-affected human liver tissue (Mean ± SD; *t*-test; *n* = 4 experiments). **i** Scheme illustrating the impact of ATP7B loss on PrP expression. In normal liver tissue, ATP7B removes excess Cu from hepatocytes (blue arrow) thereby preventing Cu-dependent induction of PrP expression. Upon ATP7B loss in WD accumulating Cu stimulates PrP expression, which in turn favors further Cu uptake/accumulation (red arrows) leading again to higher PrP expression, thus generating a vicious circle. **j** Control and MTF1-silenced ATP7B-KO HepG2 were treated with 0.5 mM CuCl$_2$ for 24 h and then prepared for qRT-PCR, which shows that Cu fails to activate *PRNP* expression in MTF1-silenced cells (Mean ± SD; two way ANOVA; *n* = 3 experiments). Scale bar: 90 μm (**g**).

## DMT1 and STEAP1 contribute to PrP-mediated Cu toxicity in ATP7B-deficient cells

PrP appears to promote Cu uptake to endocytic compartments, but any resultant cell death would require that the Cu ions are transported across membranes out of these compartments to sites that are responsible for cellular toxicity. We, therefore, evaluated if any of the validated hits from the genome-wide screen might be involved in transporting the Cu out of endosomal compartments to the cytoplasm. Two of the hits, STEAP1 and DMT1 (SLC11A2), represented putative players in this process (Fig. 5a). Cu is frequently subjected to reduction from Cu$^{2+}$ to Cu$^+$ before the ion moves across the membrane to the cytoplasm. This process could be driven either by PrP itself, which was proposed to possess Cu reductase activity[45], or by STEAP1, which supports metallor-eductase activity within a heterotrimer complex with other members of the STEAP family[46]. DMT1 has been reported to reside in the endocytic compartment and to transport Cu$^+$ across membranes[47]. Thus, DMT1 could be responsible for transporting Cu taken up by PrP across endosomal membranes to the cytoplasm. In line with this hypothesis, internalized PrP was occasionally co-detected with STEAP1 and DMT1 in endocytic structures in ATP7B-KO cells (Fig. 5b). The relatively low number of these double-positive endocytic structures would be consistent with a transient interaction between PrP and STEAP1/DMT1, and the rapid recycling of PrP back to the plasma membrane. Such recycling is believed to be triggered by conformational changes in PrP, which could occur in response to decreasing pH along the endocytic pathway and lead to the dissociation of Cu from PrP[29]. This pool of Cu released from PrP can be then handled by STEAP1 and DMT1 for transfer across the endosomal membrane to the cytoplasm (Fig. 5a).

Indeed, dose-response experiments confirmed that silencing of DMT1 and of STEAP1 reduced Cu toxicity in ATP7B-KO cells (Fig. 5c, d). Considering the functional redundancy between STEAP family members, we also investigated whether silencing STEAP2 and STEAP3 would impact the resistance of ATP7B-KO cells to Cu. We found that suppression of STEAP2 improved cell survival upon incubation with Cu (Fig. 5e), indicating a potential role of this reductase in promoting Cu toxicity in ATP7B-deficient cells. Further, by evaluating MT1X expression or CF4 fluorescence we found that suppression of STEAP1 and DMT1 significantly reduced Cu accumulation in the cells (Fig. 5f–h). Finally, TMRE labelling showed that silencing of STEAP1 and DMT1 attenuated Cu-induced loss of mitochondrial membrane potential in ATP7B-KO cells (Fig. 5i, j). This indicates that both proteins apparently act along a pathway that directs Cu from endosomes to toxicity targets in the cytoplasm and in mitochondria[4,5,31,48].

Next, we tested whether STEAP1 and DMT1 operate downstream of PrP. If this were the case, their silencing should not alter the resistance of ATP7B-KO cells to Cu toxicity conferred by PrP suppression. To investigate this, we generated ATP7B-KO HepG2 cells in which PrP expression was permanently knocked down via stable shRNA transfection (Supplementary Fig. 7a). These cells exhibit higher resistance to Cu compared to the parental ATP7B-KO cell line (Supplementary Fig. 7b). However, silencing of either STEAP1 or DMT1 did not improve

their tolerance to Cu any further (Supplementary Fig. 7c). This lack of synergy indicates that STEAP1 and DMT1 operate downstream of PrP in the pathway that directs Cu from endosomes to intracellular toxicity sites. In contrast, silencing of another screen hit, TFR2, in PrP-deficient ATP7B-KO cells increased their tolerance to Cu (Supplementary Fig. 7d), suggesting that TFR2 uses a PrP-independent pathway to counteract Cu toxicity.

## PrP suppression ameliorates the phenotype of Atp7b knockout mice

To test the pathological relevance of the PrP-mediated mechanism in vivo, we used Atp7b$^{-/-}$ mice, a well-characterized WD mouse model. Atp7b$^{-/-}$ mice recapitulate the main features of WD including accumulation of copper in the liver and extensive liver damage, which manifest in hepatitis, fibrosis, and the formation of cholangiocarcinoma[32]. We crossed Atp7b$^{-/-}$ mice with Prnp$^{-/-}$ mice and analyzed the degree of liver damage in the resulting double knockout strain (Atp7b$^{-/-}$:Prnp$^{-/-}$). Analysis of ALT (alanine aminotransferase, a serum marker of liver damage) revealed a strong increase in Atp7b$^{-/-}$ mice, while Atp7b$^{-/-}$:Prnp$^{-/-}$ animals exhibited significantly lower ALT values (Fig. 6a). This correlated with better survival of Atp7b$^{-/-}$:Prnp$^{-/-}$ mice compared to single Atp7b knockout animals (Fig. 6b). The animals were sacrificed at 28 weeks of age and the degree of liver damage was assessed by histopathology. The liver of Atp7b$^{-/-}$ mice showed severe alterations, even at the macroscopic level, with numerous abnormal nodes (Fig. 6c), while the liver of Atp7b$^{-/-}$:Prnp$^{-/-}$ animals appeared fairly regular with no nodes detected (Fig. 6d). Histological examination of the liver from Atp7b$^{-/-}$ mice revealed several WD-associated features, including the presence of giant cells, massive leukocyte infiltration, and extensive fibrosis (Fig. 6e). In contrast, the liver from Atp7b$^{-/-}$:Prnp$^{-/-}$ animals appeared fairly normal with only a few areas of leukocyte infiltration (Fig. 6f).

Further histopathological analysis revealed that the livers of Atp7b$^{-/-}$ mice contained extensive areas of proliferating biliary ducts forming tumor-like nodes (Fig. 6g; Supplementary Fig. 8a–c). These nodes were strongly stained with the cholangiocarcinoma marker CK19 (Supplementary Fig. 8d–e). Notably, immunolabeling showed the elevated PrP expression in these cholangiocarcinoma-like areas of Atp7b$^{-/-}$ mice (Fig. 6h), while no PrP signal was detected in Atp7b$^{-/-}$:Prnp$^{-/-}$ animals, as expected (Fig. 6i). These observations suggest that cells in these cholangiocarcinoma-like nodes may upregulate PrP expression to meet the elevated Cu requirements essential for tumor development and proliferation[9].

We also analyzed the ultrastructure of hepatic mitochondria in Atp7b$^{-/-}$ and Atp7b$^{-/-}$:Prnp$^{-/-}$ animals. Mitochondria in Atp7b$^{-/-}$ mice exhibited several aberrations typical of WD, including an electron-dense matrix, significant swelling of cristae, and expansion of the intermembrane space (Supplementary Fig. 9a). Moreover, these mitochondria contained patchy electron-dense particles (Supplementary Fig. 9a; inset), which correlate with mitochondrial Cu accumulation[3]. In contrast, these alterations were absent in Atp7b$^{-/-}$:Prnp$^{-/-}$ animals and the ultrastructure of their mitochondria appeared relatively normal (Supplementary Fig. 9b).

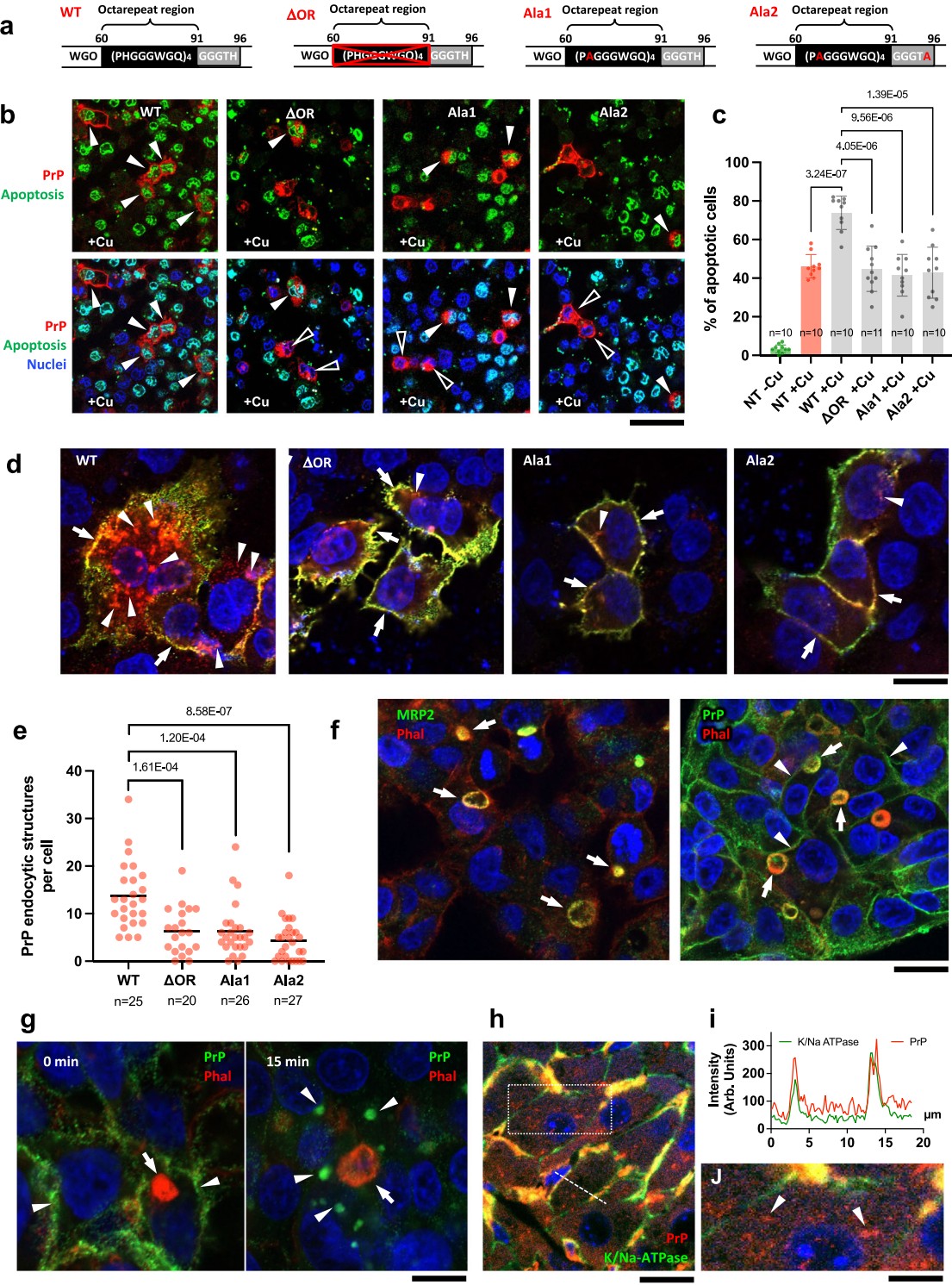

Collectively, analyses of ALT levels, liver morphology, and mitochondrial ultrastructure suggest that Prnp knockout significantly reduces liver damage in Atp7b$^{-/-}$ mice. This aligns with our in vitro findings, which show lower Cu toxicity in primary Atp7b$^{-/-}$:Prnp$^{-/-}$ hepatocytes (Supplementary Fig. 3). Unexpectedly, ICP-OES analysis did not detect a significant decrease in hepatic Cu levels in Atp7b$^{-/-}$:Prnp$^{-/-}$ mice, despite the notable improvement in phenotype (Supplementary Fig. 9c). This led us to investigate whether acute inhibition of PrP could reduce hepatic Cu uptake in Atp7b$^{-/-}$ mice. Indeed, acute suppression of PrP in the liver of Atp7b$^{-/-}$ mice via siRNA injection resulted in a significant decrease in Cu, supporting the role of PrP in hepatic Cu uptake (Supplementary Fig. 9d).

In this context, it is tempting to speculate that, in the chronic absence of PrP-mediated uptake, Cu still finds alternative pathways to enter Atp7B-deficient cells. However, these pathways result in lower toxicity, allowing cells to modulate compartmentalization and sequestration of excess Cu in a timely manner. Indeed, we observed that electron-dense particles in Atp7b$^{-/-}$:Prnp$^{-/-}$ animals accumulated inside endo-lysosomal organelles rather than mitochondria (Supplementary Fig. 9b; inset), suggesting entrapment of endocytosed Cu

**Fig. 4 | Mutation of copper binding sites reduces PrP-mediated Cu toxicity and impairs PrP endocytosis. a** Schematic representation of mutations in Cu-binding sites of PrP. **b, c** ATP7B-KO HepG2 cells were transfected with mouse WT and mutant PrP variants, then treated with Cu and stained with CellEvent reagent to reveal apoptotic nuclei. Apoptotic (arrowheads) and non-apoptotic (empty arrowheads) cells expressing PrP variants are indicated in (**b**). Quantification (**c**) indicates higher ability of PrP-WT to induce Cu-mediated apoptosis compared to non-transfected (NT) and mutant-expressing cells (Mean; one way ANOVA; $n$ = view fields). **d, e** Representative images (**d**) show surface (green) and total (red) pools of endocytosed PrP in ATP7B-KO cells stimulated with Cu for 15 min. Arrowheads in (**d**) indicate structures containing endocytosed PrP, while arrows indicate surface PrP. Quantification (**e**) shows higher number of PrP-positive endocytic structures in cells expressing WT variant compared to cells expressing the mutants (Mean; one-way ANOVA; $n$ = cell number). **f** Left panel shows polarized

cells with MRP2-positive apical/canalicular vacuoles (arrows), which contain actin reach microvilli labelled with phalloidin (Phal). Right panel shows endogenous PrP in apical vacuoles (arrows) and at the basolateral surface of polarized hepatic cells (arrowheads). **g** Polarized ATP7B-HO HepG2 cells were incubated on ice with anti-PrP antibody, which bound to PrP at the basolateral surface (arrowheads in left panel) but did not penetrate to apical vacuole (arrow). Then PrP endocytosis was stimulated with Cu and internalized PrP was revealed inside endocytic structures (arrowheads in right panel). **h** Immuno-labelling of liver sections from of Atp7b$^{-/-}$ mice shows significant overlap of PrP with basolateral marker of hepatocytes K/Na-ATPase. **i** The plot shows intensities of PrP and K/Na-ATPase signals along the dash line in (**h**). **j** The panel shows boxed area in **h** at higher magnification. Arrowheads indicate intracellular PrP-containing endosome-like structures. Scale bars: 35 μm (**b**); 18 μm (**d**); 20 μm (**f, h**), 13 μm (**g**), 7.7 μm (**j**).

within the endo-lysosomal compartment. Furthermore, investigation of Cu-transporting/binding proteins revealed changes in the expression of metallothioneins (Mt1/2) and Steap1 (Supplementary Fig. 9e). Elevated Mt1/2 levels in Atp7b$^{-/-}$:Prnp$^{-/-}$ animals should facilitate better Cu sequestration and detoxification, as reported in other Atp7b-deficient mouse strains[49]. A decrease in Steap1 expression may not favor Cu2+ reduction for subsequent transmembrane transport from endocytic organelles to the cytoplasm. As a result, some of the endocytosed copper may remain trapped in the endo-lysosomal system, while the rest might recycle back into systemic circulation and be eliminated through urinary excretion. Indeed, we found higher urinary copper levels in Atp7b$^{-/-}$:Prnp$^{-/-}$ animals compared to Atp7b$^{-/-}$ mice (Supplementary Fig. 9f). This phenotype resembles D-Penicillamine treatment, which promotes urinary Cu excretion in WD patients without reducing hepatic Cu levels[10].

In summary, our in vivo studies indicate that suppression of PrP reduces Cu toxicity in mouse model of WD.

## Discussion

Here we describe a previously uncharacterized role of PrP in Cu dyshomeostasis in Wilson disease. Our findings indicate that PrP promotes Cu toxicity by facilitating Cu uptake by ATP7B-deficient hepatic cells. Although the ability of PrP to bind Cu and operate as an endocytic Cu transporter is well documented[27,28], this protein has not been considered as an important player in WD pathogenesis and in hepatic Cu metabolism in general. This oversight could be explained in part by the higher expression of PrP in the nervous system while it is lower in other tissues, including the liver[36,37], which is primarily affected by WD. Thus, in a physiological context, it was difficult to imagine that PrP plays a significant role in hepatic Cu uptake. As a result, the Cu-binding properties/functions of PrP were mainly explored in a neurocentric context linked to prion disease.

Surprisingly, the capacity of Cu to induce *PRNP* expression in WD[23,24] has been overlooked. We found that Cu accumulation in ATP7B-deficient hepatic cells increases PrP expression through the mechanisms driven by transcription factor MTF1, which is capable of sensing elevated intracellular Cu (reviewed in ref. 50). Previous study indicated that in response to mounting Cu, MTF1 binds to metal-responsive elements (MREs) in the promoter of *PRNP* and activates its transcription[23]. We found that silencing of MTF1 leads to the failure of Cu to transactivate *PRNP* in ATP7B-KO HepG2 cells (Fig. 3j). This suggests that ability of Cu to stimulate MTF1-mediated transcription of *PRNP* leads to higher hepatic PrP expression in Wilson disease.

The combination of higher expression and ability of PrP to facilitate Cu uptake became an important factor in a tug-of-war between mechanisms facilitating or counteracting Cu toxicity. Although loss of ATP7B forces hepatic cells to defend against excess Cu by down-regulating the high-affinity transporter CTR1[39] (see also Supplementary Fig. 4j), Cu still manages to exploit lower affinity receptors/transporters and takes advantage of elevated PrP expression to evade

and poison the cell. In this context, it is worth noting that after PrP-mediated uptake, other transporters, such as DMT1, which preferentially transports iron[47], apparently have substantial affinities for Cu to propagate its toxicity. Our observations suggest that the ability of Cu to utilize iron transport and handling proteins extends beyond DMT1. We found that the iron reductases STEAP1 and 2 promote Cu toxicity in ATP7B-deficient cells. Normally, these enzymes reduce endocytosed iron, making it suitable for further transport by DMT1 from endosomes to the cytoplasm[51]. Similarly, STEAPs can reduce copper[51], facilitating its transport across the endosomal membrane in ATP7B-deficient cells (see scheme in Fig. 5a). This indicates a strong connection between copper and iron metabolic pathways during the pathogenesis of Wilson disease.

In a broader context, our findings suggest that "non-canonical" pathways of Cu uptake may be activated under certain physiological and pathological conditions. While we identified the importance of the PrP-mediated pathway in Wilson disease, a recent study reported that CD44-mediated Cu uptake plays a significant role in immunity and cancer progression[14]. Notably, CD44 has also been shown to drive iron endocytosis[52]. This points to the possibility that copper and iron might share a common pool of transport proteins in various physiological and pathological conditions. The dominant Cu uptake pathway in each situation may depend on the expression of the respective transporter. CD44 is known to be highly expressed in immune and some cancer cells[14], while elevated hepatic expression of PrP defines its importance in Wilson disease.

The same concept can be applied to understanding the mechanisms of copper toxicity. Recently, cuproptosis has been identified as a primary mechanism promoting Cu-mediated cell death in tumor cells treated with the Cu ionophore elesclomol[5]. However, the contribution of various toxicity pathways to cell death can be influenced by specific conditions of Cu supply (e.g., the presence or absence of an ionophore) or the expression of molecular players driving particular toxic processes.

Our screening identified only one cuproptosis-related hit, suggesting that the mechanisms through which Cu kills ATP7B-deficient cells are not limited to cuproptosis. Indeed, massive apoptosis was observed in ATP7B-KO cells upon Cu treatment (see Supplementary Figs. 2a, b, and 3b, d). This aligns with existing literature indicating that Cu utilizes multiple pathways to exert its toxicity[3–5]. Therefore, Cu internalized via PrP may be directed to different intracellular toxicity routes. Our data suggest that some of these routes converge on mitochondria, as PrP suppression significantly reduces mitochondrial damage in both cell and animal models of WD.

How important is the contribution of PrP in the pathogenesis of WD? We found that genetic PrP suppression significantly ameliorates the phenotype in Atp7b$^{-/-}$ mice, substantially reducing liver damage. This indicates that a significant part of hepatic Cu toxicity relies on PrP, whose activity/expression may influence the severity of WD. Indeed, expression of different PrP variants has been linked to either earlier

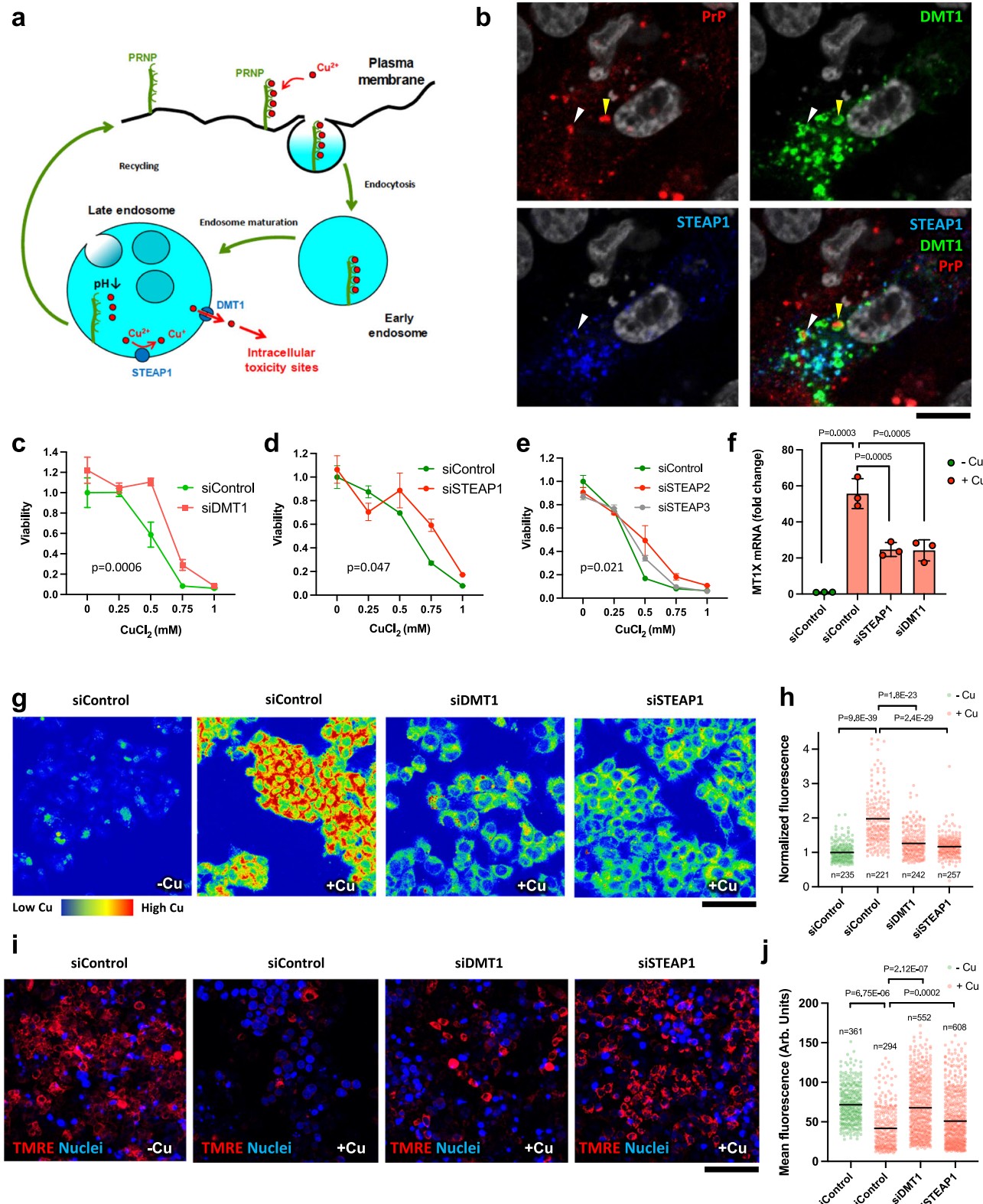

onset of the disease or to the severity of its manifestations[20–22,33]. Clinical studies suggest that *PRNP* polymorphism at codon 129 may influence the severity of WD. These studies, however, do not provide a definitive conclusion regarding whether the presence of a specific amino acid residue at this position in PrP plays a protective or toxic role in Wilson disease. Homozygosity for the M129 variant of PrP has been associated with either a later onset of the disease[33] or with

significantly more severe neurological symptoms in patients with WD[20]. We compared homologous mouse variants M128 and V128 in PrP for their ability to promote Cu toxicity in ATP7B-KO cells but did not observe any differences (Supplementary Fig. 4). Of particular interest was another clinical observation that a WD patient with the G54S *PRNP* variant presented with very severe symptoms, while his older sister, who carried the same ATP7B genotype but was clinically

**Fig. 5 | DMT1 and STEAP1 promote Cu toxicity in ATP7B-KO cells. a** The scheme shows hypothetical pathway of PrP-mediated Cu uptake in ATP7B-deficient cells. **b** ATP7B-KO cells were transfected with DMT1-GFP and STEAP1-Myc DNA constructs. The cells were incubated on ice with anti-PrP antibody. Then PrP endocytosis was stimulated by incubating the cells at 37 °C in medium containing 200 μM $CuSO_4$ for 15 min. Internalized PrP was revealed inside endocytic structures, some of which contained DMT1 (yellow arrowheads) or both DMT1 and STEAP1 (white arrowheads). ATP7B-KO cells were incubated with control (siControl), DMT1-specific (**c**), or STEAPs-specific (**d, e**) siRNAs and exposed to $CuCl_2$ for 24 h. MTT assay indicates that DMT1- or STEAP1- or STEAP2-silenced cells have higher viability upon exposure to Cu (Mean ± SEM; two-way ANOVA; $n = 3$ experiments). **f** ATP7B-KO cells were treated with control, DMT1-specific, or STEAP1-specific siRNAs and exposed to 0.5 mM $CuCl_2$ for 24 h. The cells were then prepared for qRT-PCR to evaluate MT1X expression, which was significantly reduced by DMT1- and STEAP1-specific siRNAs (Mean ± SD; one-way ANOVA; $n = 3$ experiments). **g, h** ATP7B-KO cells were incubated with control, DMT1-specific, or STEAP1-specific siRNAs and exposed to 0.5 mM $CuCl_2$ for 24 h. Then the cells were stained with fluorescent copper sensor CF4. Representative false color images (**g**) and quantification (**h**) demonstrate reduction of CF4 signal in DMT1- and STEAP1-silenced cells upon Cu treatment (Mean ± SD; one-way ANOVA; $n = $ cell number). **i, j** ATP7B-KO cells were treated as described in (**g**). Then the cells were labelled with mitochondrial membrane potential dye TMRE. Representative micrographs (**i**) and their quantification (**j**) show that DMT1 or STEAP1 suppression increases TMRE signal in Cu-treated cells (Mean ± SD; one way ANOVA; $n = $ cell number) indicating improvement of mitochondrial function. Scale bars: 8 μm (**a**); 100 μm (**g**); 220 μm (**i**).

asymptomatic, lacked this variant allele[22]. Consistent with this, we found that the expression of the mouse G53S homolog of this PrP variant enhanced Cu toxicity in ATP7B-KO cells (Supplementary Fig. 4). This indicates that *PRNP* may function as an important gene modifier in WD[21], and the Cu-binding and transporting properties of WD-modulating variants of PrP warrant further investigation.

Our data also suggest that PrP has to be considered in the broader context of Cu metabolism and Cu-related disorders. On one hand, higher expression of PrP might help to satisfy a metabolic demand for extra copper in cells or tissues with elevated Cu consumption[26]. On the other hand, high PrP expression could promote abnormal accumulation of Cu, which fuels numerous processes favoring malignant transformation of cells and tissues[9]. In agreement with this, we observed the highest expression of PrP in cholangiocarcinoma nodes in the liver of Atp7b$^{-/-}$ mice, whereas PrP depletion almost completely prevented tumor growth. High PrP expression has been also reported as an unfavorable factor for overall survival of patients with hepatocellular carcinoma[53]. Thus, a growing body of evidence indicates that PrP is tightly integrated in the mechanisms regulating Cu transport and distribution in health and disease. It may be worth exploring the role of PrP in other genetic disorders causing alterations in Cu metabolisms (such as MEDNIK syndrome or AP1B1 deficiency) and in cancer. A number of PrP suppression strategies are being explored for the therapy of prion disease[54]. Our data indicating that PrP plays a role in mediating copper toxicity in WD warrant future studies to evaluate the therapeutic potential of PrP suppression in this and other disorders of copper metabolism.

## Methods

### Ethics statement
Research reported in this study complies with all relevant ethical regulations and was authorized by the Italian Ministry of Health (Authorization n° 913/2021-PR) and approved by ethical boards of TIGEM or Institute of Experimental Medicine. All animal experiments were performed in line with the ARRIVE guidelines.

### Antibodies and plasmids
The followed antibodies were used: mouse monoclonal anti-PrP 3F4[55] (dilution: 1:100 for IF), mouse monoclonal anti-PrP 6D12 (Wageningen University and Research, dilution: 1:100 for IF), mouse monoclonal anti-PrP 12B2, 100B3, 94B4 (Wageningen University and Research, dilution: 1:100 for IF, 1:1,000 for WB; 1:100 for IHC); Alexa Fluor 568 Phalloidin (Thermo Fisher, A12380, dilution: 1:600 for IF); mouse monoclonal anti-MRP2 (Enzo Lifescience, ALX-801-016-C250, dilution: 1:50 for IF); mouse monoclonal anti-PrP SAF32 (Bertin Bioreagent, A03202, dilution: 1: 50 for IF), rabbit anti-Myc tag (Millipore, 06549, dilution 1:100), mouse monoclonal anti-K/Na-ATPase (Abcam, ab7671, dilution 1:100), mouse anti-GAPDH (Santa Cruz Biotechnology, sc-32233, dilution 1:1,000 for WB), rabbit monoclonal anti-ATP7B (Abcam, Ab 124973, dilution 1:1,000 for WB) or mouse monoclonal anti-Vinculin (Sigma–Aldrich, V9264, dilution 1:5,000 for WB).

pCDNA3.1(+) plasmids encoding mouse PrP WT (M128) or M128V have been described[56]. Mouse PrP cDNA constructs carrying a deletion of the OR region (ΔOR), or in which the histidines in the OR (Ala1) or in the OR and non-OR region (Ala2) was substituted with alanines, or in which the glycine in position 53 was substituted with a serine (G53S), were synthesized by GeneArt (Thermo Fisher) and inserted between the HindIII and BamHI restriction sites of pCDNA3.1(+). DMT1-GFP cDNA was kindly provided by Prof. Izumi Yanatori (Nagoya University Graduate School of Medicine, Japan). Myc-tagged STEAP1 cDNA was purchased from OriGene.

### Cell culture and transfection
Parental or ATP7B-knockout (KO) HepG2 cells[12] were grown in RPMI medium supplemented with 10% fetal calf serum (Euroclone), 1% L-glutamine (Euroclone), and 1% penicillin/streptomycin (Euroclone). The ATP7B-KO cell line with stable PRNP knockdown (ATP7B-KO:PRNP-KD) was generated by infecting ATP7B-KO cells with pLKO1-puro-CMV-tGFP shPRNP-lentiviral particles (Mission lentiviral transduction particles, Sigma–Aldrich) and upon selection with puromycin. Polarization of ATP7B-KO cells was achieved upon treatment with 20 ng/ml Oncostatin M (Sigma–Aldrich) in media for 72 h. The degree of cell polarization was assessed based on the formation of apical (canalicular) vacuoles, which were revealed using MRP2 or Phalloidin staining (see below). For cDNA plasmid transfection, cells were plated in a 24-well plate and transfected using LipoD293 (Signa Gen Laboratories) with a total of 500 ng of plasmid DNA per well.

Primary hepatocytes cells were isolated from livers of 6-week-old Atp7b$^{-/-}$, Prp$^{-/-}$; Atp7b$^{-/-}$ or control HZ Atp7b mice by a modified protocol based on Pronase/collagenase digestion. In brief, mouse livers were perfused through the inferior vena cava with EGTA solution followed by enzymatic digestion with Pronase (Sigma–Aldrich) and then collagenase type D (Roche Applied Science). Next, livers were harvested, and liver cells were disassociated by digestion with Pronase/collagenase solution and filtered through a nylon filter (Corning) to remove undigested tissues and debris. The resulting cell suspension was centrifuged and washed three times. The final cell suspension was plated in a 24 well plate on Bovine collagen substrate (Advanced BioMatrix) In Hepatozyme (Gibco) media supplemented with 1% Pen Strep, and 5% ITS solutions (Thermofisher). The day after, dead cells were removed through medium washes prior being treated with different concentrations of $CuCl_2$ to assess resistance to copper by Live/Dead or apoptotic assays (see below).

### RNA interference and drug treatments
Small interfering RNAs (siRNAs) were custom-designed and purchased from Sigma–Aldrich (Supplementary Dataset 4). Scrambled ON-TARGETplus Non-targeting Control siRNAs were used as a control (Dharmacon). HepG2 cells were transfected with individual or pooled siRNAs using Darmafect4 (Dharmacon) according to the manufacturer's instructions. Silencing efficiency was assayed using qRT-PCR. Efficiency of the target gene silencing was confirmed by qRT-PCR.

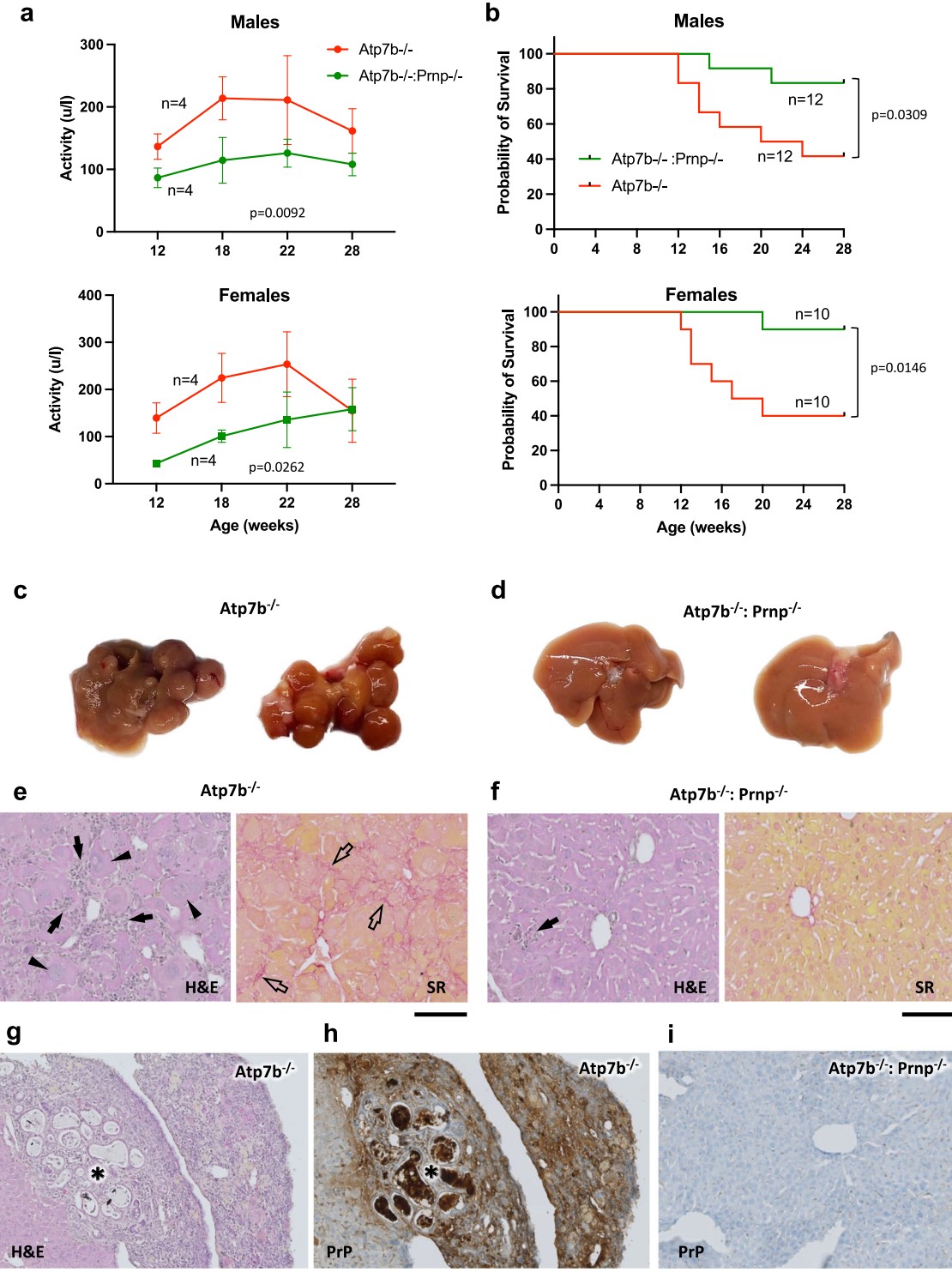

**Fig. 6 | Genetic suppression of *Prnp* improves phenotype in Atp7b⁻/⁻ mice. a** The graph shows dynamics of activity of serum ALT in Atp7b⁻/⁻ or Atp7b⁻/⁻:Prnp⁻/⁻ mice (Mean ± SEM; two-way ANOVA; *n* = animals). **b** The plot demonstrates better survival of Atp7b⁻/⁻:Prnp⁻/⁻ mice compared to Atp7b⁻/⁻ animals (Log-rank Mantel-Cox test; *n* = animals). **c, d** Macroscopic images demonstrate multiple abnormal nodes in the livers of Atp7b⁻/⁻ mice (**d**) while livers of Atp7b⁻/⁻:Prnp⁻/⁻ animals exhibit fairly normal morphology. Sections of liver from Atp7b⁻/⁻ (**e**) or Atp7b⁻/⁻:Prnp⁻/⁻ (**f**) mice were stained with either hematoxylin and eosin (H&E) to evaluate overall morphology or with Sirius Red (SR) to reveal fibrotic areas (visible in red color). Arrowheads in (**e**) indicate abnormal giant hepatocytes. Arrows in (**e** and **f**) show leukocyte infiltrations. Empty arrows in (**e**) indicate fibrotic regions. **g, h** Sections of liver from Atp7b⁻/⁻ mouse were obtained from the same area of cholangiocarcinoma node (asterisk) containing proliferating biliary ducts. The sections then were either stained with hematoxylin and eosin (**g**) or immuno-HRP labelled with antibody against PrP (**h**), which reveals higher expression of PrP in the tumor (asterisk) compared to flanking parenchyma. **i** Section of liver from Atp7b⁻/⁻:Prnp⁻/⁻ was immuno-HRP labelled with antibody against PrP, which did not reveal any PrP expression. Micrographs in (**e–i**) represent results reproduced in 3 different animals with similar outcomes. Scale bars: 100 μm (**e**, **f**); 160 μm (**g–i**).

Treatment with SM875 (10 μM; ref. 35) or with tetracationic zinc (Zn(II)-BnPyP), copper (Cu(II)-BnPyP), iron (Fe(III)-BnPyP), and empty apo-BnPyP porphyrins at a concentration of 5 μM each[34] was performed in ATP7B-KO cells for 24 h followed by the addition of different concentrations of $CuCl_2$ for 24 h. Cells were then analyzed with the MTT viability assay, the live/dead assay, or processed for protein or RNA isolation as described below.

## MTT and live/dead viability assays

Cell viability after treatment was determined by measuring the ability to reduce the tetrazolium salt (MTT) (Life Technologies) to a formazan using a previously described procedure[15]. For the live/dead cytotoxicity assay, the cells were washed in PBS and incubated with the Live/ Dead reagent (Life Technologies) for 45 min following the manufacturer's instructions. Labelled cells were viewed and analyzed using the Opera High Content (HC) screening system (Perkin Elmer) with FITC and RFP filters to count live/dead cells in the treated and control specimens. Quantification was done in at least 5 fields for each condition and the proportion of live cells was calculated as a percentage of total cells in the specimen.

For specific staining of the dead cells after expression of WD-associated G53S or M128V PrP variand, control, and treated cells were washed in warm media and incubated for 1 h at 37 °C in the incubator with BOBO3 (Life Technologies) diluted 1:1000 in the media. After incubation, cells were washed in PBS and immunolabelled for PrP. Images of the labelled cells were acquired at Nikon AXR confocal microscope. Quantification was done in at least 10 fields for each condition and the proportion of dead cells was calculated as a percentage of total cells expressing the WT or WD-associated variants.

## Apoptotic assay

To analyze cell apoptosis, the cells were incubated with 6 mM Cell Event Caspase-3/7 Green Detection Reagent (Life Technologies) for 30 min, then fixed and labeled with DAPI to counterstain nuclei. After activation of Caspase-3/7, the apoptotic cell nuclei stain green due to the binding of the Cell Event dye to DNA. Images of labeled cells were acquired at Zeiss LSM700 confocal stage with a 20× objective, appropriate laser excitation lines, and filter sets. Quantification of apoptotic cells was performed in 10 fields for each condition and the proportion of apoptotic cells was calculated as a percentage of total cells in the specimen.

## Genome-wide shRNA screenings in ATP7B-KO cells

**Cell transduction with a lentiviral shRNA screening library and gDNA isolation.** To identify genes whose suppression reduces Cu toxicity in ATP7B-KO cells, the Decode™ Pooled Lentiviral shRNA Screening Library (Cat. #RHS6083, Dharmacon) was used. This library targets 18,205 human genes and contains 10 pools of 9570 GIPZ short hairpin RNAs (shRNAs) packaged into high-titer lentiviral particles. Each target gene has 4–7 corresponding shRNAs distributed across the different pools.

Transduction of ATP7B-KO cells with each pool was performed according to the manufacturer's instructions with a lentiviral MOI of 0.3 to ensure the integration of not more than one shRNA per cell. The number of cells seeded for transduction was calculated to have at least 1000 copies for each shRNA in the transduced cell population. Viral particles diluted in 9 ml of transduction media (DMEM with no serum or antibiotics) and 10 mg/ml polybrene were added to the cells, which were further propagated with 2 mg/ml of puromycin for 96 h to select the population with stable shRNA integration. For each shRNA pool, the transduced cells were expanded to at least ten 100 mm plates to maintain shRNA copy representation. The cells were then divided into reference group, which was not treated, and treated group, which was incubated with 0.5 mM $CuCl_2$ for 24 h. The cells from both reference and treated groups were collected in PBS and subjected to apoptotic cell removal with the Apoptotic Cell Isolation Kit (Novus Biologicals). Cell pellets obtained after the apoptotic cell removal were frozen until processing for genomic DNA (gDNA) isolation. gDNA was isolated from transduced cells using Quick-DNA Miniprep Plus kit (Zymo Research) according to the manufacturer's procedure. For each library pool, 3 gDNA specimens from the reference group and 3 gDNA specimens from the treated group were collected for PCR amplification. Each specimen contained at least 6.6 ug of gDNA to maintain representation at 1,000 copies per shRNA.

**PCR amplification.** gDNA specimens were further PCR amplified with index primers recognizing bar codes, which allows each shRNA copy to be assigned to a specific specimen during the demultiplexing step of sequencing data analysis. Amplification was performed using the manufacturer's protocols and Decode PCR primers (Darmacon) provided together with the shRNA library. This protocol is optimized to avoid nonlinear amplification of shRNA template copies present in a specimen[57]. For each specimen, eight technical replicates (each containing 825 ng of gDNA) were amplified with Decode PCR primers using 96-well plates. In each well, a 50 ml reaction mix contained 200 mM dNTP, 0.5 M betaine, and 0.08 U/ml of Phusion Hot Start II DNA Polymerase (Sigma–Aldrich). Individual reactions from replicates belonging to the same specimen were pooled together and the presence of a 660 base pair amplicon for each specimen was confirmed by agarose gel electrophoresis. DNA amplicons were purified using the Quick DNA MiniPrep Plus kit (Zymo Research) and their yield and quality were evaluated using a Nanodrop 1000 spectrophotometer (Thermo Fisher).

**Library preparation and next-generation sequencing (NGS).** For preparation of the library containing single samples, we followed the manufacturer's instructions (Dharmacon). Purified DNA samples were validated and quantified by microfluidic analysis using the Bioanalyzer High Sensitivity DNA Assay kit (Agilent Technologies) and the 2100 Bioanalyzer with 2100 Expert Software. All samples were pooled with a 10% phiX spike-in. The library was sequenced using the Next Seq500 system performing single read (SR) runs covering at least 75 nt. (Illumina Inc., San Diego, CA, USA).

**Data analysis.** To analyze the NGS datasets, we first performed a demultiplexing step for the high-throughput sequencing output with the bcl2fastq tool (ver. 2.20.0.422) (https://scicrunch.org/resolver/ SCR_015058), obtaining one FASTQ file for each sample. To measure the relative quantity of each shRNA in a sample, we aligned the NGS reads to the construct FASTA reference files, provided with the Dharmacon library, using Bowtie2 (ver. 2.4.2)[58]. The Bowtie2 output was converted into a single raw count matrix for subsequent analysis using the Python (ver. 2.7.18) script cpuntBowtie2Hits.py, which was provided by Dharmacon. Finally, we normalized raw counts into counts per million (CPM), computed the differential expression analysis between the two conditions of the experimental design, and annotated each shRNA with the corresponding gene. We performed these steps in R environment (ver. 4.2.3), using NOISeq package (ver. 2.42.0) for the normalization[59,60], and edgeR package (ver. 3.40.2) for the identification of shRNAs differentially represented in reference and treated cell populations[61].

**Hit gene selection.** The gene hits were selected according to the following criteria. Each hit gene must have at least two corresponding shRNAs enriched in the treated specimen, with a statistically significant false discovery rate (FDR) < 0.05. These enriched shRNAs must belong to different pools of the library, and at least one of them has to be enriched more than twice (LogFC ≥ 1). Finally, no significantly depleted shRNA should correspond to the hit gene.

**Vinblastine screening.** The genome-wide screen was repeated with the same pooled shRNA lentiviral library using the proapoptotic drug vinblastine (60 μM for 24 h) instead of CuCl$_2$ as a selective pressure treatment. The hit lists from the CuCl$_2$ and vinblastine screens were compared and common hits were excluded from further analyses. This is because their suppression is likely to inhibit generic cell death mechanisms rather than specifically counteracting copper toxicity in ATP7B-KO cells.

## Gene ontology enrichment analysis
The list of the hit genes was subjected to gene ontology (GO) enrichment analysis. The GO analysis was conducted using DAVID Bioinformatic Resources[62] and output was restricted to Biological Process Functional Annotation Terms (BP_FAT). The threshold for statistical significance of GO analysis was FDR < 0.1.

## Validation of target genes in secondary screenings
A curated list of target genes (Supplementary Dataset 3), which emerged from the shRNA screen and subsequent GO analysis, was used for further validation with siRNAs selected from the Human Druggable Genome siRNA Library (Dharmacon). ATP7B-KO cells were reverse transfected with siRNAs using RNAiMax (Thermo Fisher) in 384-well plates. Each siRNA (at 50 uM concentration) was tested alone for 48 h to evaluate toxicity and, in a parallel set of plates, each siRNA was combined with 0.5 mM CuCl$_2$ for a further 24 h. The STAR-let liquid handling system (Hamilton, Reno, NV, USA) was used to prepare the assay plates. Wells containing scramble siRNAs were used as controls. After treatment, the cell viability was evaluated using the MTT assay as described above. siRNAs that increased cell viability only in combination with CuCl$_2$ were considered as validated and advanced to subsequent screening with the Live/Dead fluorescent reagent using the Opera HCS system (see above). Genes whose siRNA reduced copper toxicity in both MTT and Live/Dead screening were considered for further investigation.

## RNA extraction and Real-time PCR
For gene expression analyses, total RNA was extracted from cells or liver tissues using Trizol and Tissue Lyser for 3 m in while for cells the RNeasy Plus Mini Kit (Qiagen) was used. Reverse transcription was performed using QuantiTect Rev Transcription Kit (Qiagen) from 1 mg of RNA according to the manufacturer's protocol. qPCR reactions were performed using SYBR Green Master Mix and run on a LightCycler480 system (Roche). Data were analysed using Light Cycler 480 software, version 1.5 (Roche Applied Science). Primer sequences are listed in Supplementary Dataset 5.

## Western blot analysis
For protein analysis, cells or pieces of liver tissue were washed in PBS and lysed in Lysis Buffer (20 mM Tris-HCl pH8, 1 mM NaCl, 0.5% NP40, 0.5% Triton-X-100, 10% glycerol) supplemented with a protease inhibitor cocktail (Sigma–Aldrich). Lysates were incubated for 30 min on ice and centrifuged for 20 min. Tissue samples were snap frozen and homogenized in Lysis Buffer with protease inhibitor using Tissue Lyser (Biorad) for 3 min. Supernatants were collected and protein content was determined using the BCA assay (Thermo Fisher Scientific). Protein samples were separated by SDS-PAGE using 4%–12% polyacrylamide gels (Bio-Rad). Primary antibody mouse anti-PrP 12B2 (Wageningen University and Research) and mouse anti-GAPDH (Santa Cruz Biotechnology; Cat#sc-32233), anti-rabbit ATP7B (Abcam) or anti-Vinculin (Sigma–Aldrich) were diluted in 5% milk in TBS-T (0.8% NaCl, 0.02% KCl, 0.3% Tris-base, 0.1% Tween20) (Bio-Rad). Proteins of interest were detected with horseradish peroxidase (HRP)-conjugated goat anti-mouse IgG antibody (GE Healthcare). Peroxidase substrate was provided using the ECL Western Blotting Substrate kit (Pierce).

## TMRE labeling
To assess the impact of Cu on mitochondria, control, and treated cells were incubated with 250 nM of the mitochondrial membrane potential dye TMRE (Thermo Fisher Scientific) for 30 min, and then images were examined under a Zeiss LSM 700 confocal microscope. Images were captured using 20× objective and appropriate excitation laser lines and filter sets. The mean intensity of the TMRE signal per cell was quantified in captured images using ImageJ software (National Institutes of Health). At least 200 cells were quantified for each condition.

## Detection of copper with a fluorescent sensor
Fluorescent Copper Fluor-4 (CF4) sensor, which reveals bioavailable Cu$^+$ pools[63], was kindly provided by Prof. Christopher J. Chang (University of California, Berkeley). Treated and control cells grown in Lab-Tek™ II Chambered Coverglass (Nunc) were rinsed with PBS and then incubated in serum-free media supplemented with 1.5 μM CF4 for 15 min. After incubation, cells were washed with PBS and left in warm serum-free media for imaging at a Zeiss LSM700 confocal microscope. Image acquisition and quantification of CF4 signal was conducted as described above for TMRE.

## Immunofluorescence, PrP endocytosis, and confocal microscopy
Immunofluorescent (IF) labeling of ATP7B-KO HepG2 cells was conducted as described in ref. 64 To investigate how the expression of PrP mutants impacted on Cu-mediated cell death, IF labelling of the mutant-expressing cells was combined with the Cell Event apoptotic assay. ATP7B-KO cells were transfected with WT, ΔOR, Ala1, or Ala2 PrP plasmids and after 48 h treated with 0.5 mM CuCl2 (Sigma–Aldrich) for 24 h. Before fixation, the cells were incubated with Cell Event reagent to identify apoptotic cells. Expression of transfected PrP plasmids in fixed cells was revealed with the 6D12 antibody (Wageningen University and Research). DAPI (Thermo Fisher) was used to counterstain cell nuclei. Quantification of apoptotic cells in transfected and non-transfected cell populations was done as described above.

The endocytosis of PrP variants in ATP7B-KO cells was evaluated in pulse-chase experiments as previously described[44]. Cells transfected with WT or mutant PrP constructs were incubated with the anti-PrP 6D12 antibody on ice. The cells were then washed and placed at 37 °C in medium containing 200 μM CuSO$_4$ for 15 min to stimulate endocytosis of PrP and the bound antibody. The cells were fixed with 4% paraformaldehyde and the surface pool of PrP was revealed using an anti-mouse Alexa488-cojugated secondary antibody. Then the cells were permeabilized with blocking/permeabilizing solution (50 mM NH$_4$Cl, 0.1% Saponin, 1% BSA in PBS), and an Alexa568-conjugated secondary antibody was used to label endocytosed anti-PrP antibody bound to its ligand. Cell images were acquired using a Zeiss LSM800 confocal microscope and endocytic PrP structures were identified as containing Alexa568 only and quantified as the number per cell using ImageJ software.

To assess Cu-dependent endocytosis of endogenous PrP, ATP7B-KO cells were pulsed with the 100B3 antibody (Wageningen University and Research) on ice and then incubated with CuSO$_4$ as described above. Subsequently, the surface pool of the bound antibody was stripped by acid wash. The cells were fixed, permeabilized and the endocytosed pool of PrP-bound antibody was revealed with Alexa568-cojugated secondary antibody. The presence of DMT1 and STEAP1 in PrP-containing endocytic structures in ATP7B-KO cells co-transfected with DMT1-GFP and STEAP1-Myc constructs was investigated using an LSM700 confocal microscope.

Endogenous PrP in polarized ATP7B-KO cells was labeled with the SAF32 antibody (Bertin Bioreagent) and visualized using the Zeiss LSM800 confocal system. PrP distribution along the basolateral (sinusoidal) or apical (canalicular) domains was assessed on the basis of its presence or absence in the apical vacuoles, which were labelled

with either an anti-MRP2 antibody (Sigma–Aldrich) or Alexa568-conjugate Phalloidin (Thermo Fisher). Endocytosis of endogenous PrP in polarized ATP7B-KO cells was evaluated with the SAF32 anti-PrP antibody using pulse/chase protocol combined with acid wash as described above.

All experiments addressing localization and endocytosis of endogenous PrP in ATP7B-KO cells were analyzed at LSM700 or LSM800 confocal microscopes (Zeiss). The images were obtained using 63× immersion objectives, appropriate excitation laser lines, and filters. The pinhole diameter was adjusted to 1 Airy unit for each emission channel.

### Electron microscopy
Electron microscopy was performed as previously described[13]. In brief, after 24 h incubation in 0.5 mM $CuCl_2$, control, and PrP-depleted ATP7B-KO cells we fixed using 1% glutaraldehyde prepared in 0.2 M HEPES buffer for 30 min. Fixed cells were scraped, pelleted, post-fixed in $OsO_4$ and uranyl acetate, dehydrated, embedded in Epon, and polymerized at 60 °C for 72 h. For each sample, thin sections were cut using a Leica EM UC7 ultramicrotome (Leica Microsystems). EM images were acquired from thin sections using a FEI Tecnai-12 electron microscope (FEI) equipped with a VELETTA CCD digital camera (Soft Imaging Systems). Abnormal mitochondria were identified based on specific ultrastructural features[13]. The proportion of abnormal mitochondria was quantified as a percentage of total mitochondria in each analyzed cell using iTEM software (Olympus).

### Animal studies
Mouse procedures were carried out in accordance with the regulations and authorized by the Italian Ministry of Health (Authorization n° 913/2021-PR). $Atp7b^{-/-}$ and $Prnp^{-/-}$ mouse strains were maintained on C57BL/6 × 129S6/SvEv and C57BL/6 background, respectively. To achieve genetic suppression of the PrP protein in an $Atp7b^{-/-}$ background, we crossed $Atp7b^{-/-}$ and $Prnp^{-/-}$ mice to generate a new $Atp7b^{-/-}$:$Prnp^{-/-}$ strain. All animals were housed under specific pathogen-free conditions at the TIGEM animal facility (Pozzuoli, Italy) with 12-h light/dark cycles and received food and water *ad libitum* with Cu in diet at 8 µg/kg concentration (Cat, #FRDBA 10-10; Ssniff Spezialdiäten GmbH, Germany). Animal welfare was constantly monitored, and euthanasia was performed according to ethical regulations. For genotyping, DNA from toe clipping of littermate animals was extracted using a house-made Lysis buffer (0.1 M Tris-HCl pH 7.5, 0.1 M EDTA, 0.1 M NaCl, 1% SDS), and the PCR was performed using primers described in Supplementary Dataset 5 and the products were analyzed on an agarose gel. Both male and female mice were used for experimental procedures. Blood was collected from the submandibular (facial) vein at 12, 18, 22, and 28 weeks after birth. The serum was collected and stored at −80 °C until further analysis. Measurements of serum ALT levels were performed using Scil Vitro Vet system (Scilvet). Urine and feces of mice were collected from 28-week-old animals using the metabolic cages for 24 h and subjected to Cu analysis using ICP-OES. The animals were sacrificed at 28 weeks of age to collect liver tissue for Western blot, qRT-PCR, histological analysis, immune labelling, and ICP-OES analysis.

For acute in-vivo PrP silencing in $Atp7b^{-/-}$ mice, the Invivofectamine 3.0 system (ThermoFisher) was used. The siRNA duplex solution was prepared and diluted to 2.4 mg/ml according to the manufacturer's instructions. Preparation of the final injection solution was also in accordance with the Thermofisher protocol. Briefly, siRNA (PrP Ambion pre-designed siRNA #4457308) was mixed with the complexation buffer and then the Invivofectamine 3.0 reagent (#IVF3005, ThermoFisher). Upon vortexing, the solution was incubated at 50 °C for 30 min and then diluted in PBS. Up to 200 µl was injected in the lateral caudal vein at a final concentration of 1 mg/kg weekly for a 6-week period. The animals were then sacrificed, and livers were collected for ICP-OES analysis while Prnp silencing was confirmed by qRT-PCR.

### Liver histological staining and immunolabelling
Pieces of liver tissue from postmortem PBS-perfused mice (at least 3 animals per group) were fixed in 4% paraformaldehyde for 12 h, stored in 70% ethanol, and embedded in paraffin blocks and sectioned using an RM2165 microtome (Leica Microsystems).

For assessment of tissue morphology, sections were stained with hematoxylin and eosin (H&E). To this end, the sections were rehydrated and stained in Mayer's hematoxylin (Bio-Optica) for 4 min. After two washes in tap water for 5 min, sections were incubated in a solution of 0.1% ammonia water (1 mL ammonium hydroxide in 1 L distilled water) for 1 min, washed again in tap water for 5 min, and counterstained in eosin y solution (Sigma–Aldrich) for 30 s.

To evaluate the extent of fibrosis, liver sections were rehydrated and stained for 1 h in picro Sirius red solution (0.1% Sirius red in saturated aqueous solution of picric acid).

For immunoperoxidase labelling of PrP in the liver, paraffin sections were rehydrated and permeabilized in PBS/0.2%-Triton (Sigma–Aldrich) for 20 min. Antigen unmasking was performed in 0.01 M citrate buffer in a microwave oven. Endogenous peroxidase activity was blocked by incubating sections in methanol/1.5% $H_2O_2$ (Sigma–Aldrich) for 30 min and incubated with blocking solution (3% BSA [Sigma–Aldrich], 5% donkey serum [Millipore], 1.5% horse serum [Vector Laboratories] 20 mM $MgCl_2$, 0.3% Triton [Sigma–Aldrich] in PBS) for 1 h. Sections were incubated with mouse anti-PrP antibody (100B3, Wageningen University and Research) overnight at 4 °C and then with biotinylated horse anti-mouse/rabbit IgG (Vector Laboratories) for 1 h. Biotin/avidin-HRP signal amplification was achieved using the ABCE lite Kit (Vector Laboratories) according to the manufacturer's instructions. 3,3'-diaminobenzidine (Vector Laboratories) was used as the peroxidase substrate. Mayer's hematoxylin (Bio-Optica) was used for counterstaining. Sections were dehydrated and mounted in Vectashield (Vector Laboratories). Image capture from sections stained with H&E, Sirius Red, or anti-PrP was performed using AxioScan.Z1 slide scanner system (Zeiss). The whole digital slides were viewed by ZEISS's Zen Blue software.

For colocalization experiments, immuno-labeling of liver sections from $Atp7b^{-/-}$ mice was performed in paraffin sections using the VENTANA BenchMark Ultra automated staining instrument (Ventana Medical Systems, Roche). Sections were deparaffinized using EZ Prep solution (950–102) for 16 min at 72 °C. Epitope retrieval was accomplished with CC1 solution (950–224) at a high temperature (95 °C) for a period that is suitable for liver tissue. Antibodies against PrP (100B3, Wageningen University and Research) and K/Na-ATPase (Abcam) were titered with a blocking solution into user-fillable dispensers for use on the automated stainer. For fluorescent detection of primary antibodies, slides were developed using the DISCOVERY FAM Kit (760–243) for 8 min. Slides were then counterstained with DISCOVERY QD DAPI (760–4196) for 8 min. To assess coincidence of PrP and K/Na-ATPase staining, the sections were visualized on a Zeiss LSM800 confocal microscope using a 63× oil immersion objective, appropriate laser excitation lines and filter sets. Obtained images were analyzed using ZEISS's Zen Blue software.

### Copper measurement
Copper levels in tissue homogenates, feces, urine, or in cell lysates were analyzed by ICP-OES (Ciros Vision, SPECTRO Analytical Instruments GmbH) after treatment of samples with 65% nitric acid (Merck) as previously described[3,48]. The concentration of copper in different samples was expressed as follows: in tissue homogenates, as µg/g wet weight; in feces, as µg/g dry weight; in urine, as µg/L; and in cells, as ng/µg protein.

### Statistics and reproducibility

Statistical analyses were performed using GraphPad Prism software (Version 10). Statistical significance was determined using *t*-test for pairwise comparison or using one-way or two-way ANOVA for multiple comparisons followed by Tukey's or Dunnett's post hoc test. The same software was used to generate survival curves, whose statistical comparison was conducted using Log-Rank (Mantel-Cox) test. Experimental group sizes (n) and *p* value significance levels are reported in the figures. Data are shown as average ± SD or average ± SE, and *p* values less than 0.05 were considered to be statistically significant. All experiments (except genome-wide screening) were repeated independently three times with similar results.

### Reporting summary

Further information on research design is available in the Nature Portfolio Reporting Summary linked to this article.

### Data availability

Sequencing data were deposited to Gene Expression Omnibus (GEO repository) with following accession numbers: GSE277865 (https://www.ncbi.nlm.nih.gov/geo/query/acc.cgi?acc=GSE277865), GSE260454 (https://www.ncbi.nlm.nih.gov/geo/query/acc.cgi?acc=GSE260454). Results of NGS and Gene Ontology Enrichment analyses are available in the supplementary datasets. Light and electron microscopy images supporting the microscopy data shown in the figures have been deposited in a publicly accessible Zenodo repository and are available at the following links: https://doi.org/10.5281/zenodo.14500237 and https://noprofittigem.sharepoint.com/sites/PolishchukLabPapers. Full uncropped images of Western blots are available in the supplementary information file. All statistical data presented in the graphs are included in the Source Data file. Source data are provided with this paper.

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

## Acknowledgements

This study received support from following projects/agencies/organizations: #NEXTGENERATIONEU (NGEU) funded by the Ministry of University and Research (MUR), National Recovery and Resilience Plan (NRRP), project MNESYS (PE0000006) – A Multiscale integrated approach to the study of the nervous system in health and disease (DN. 1553 11.10.2022); Telethon Italy (grant #TGM22CBDM05 to RSP, #GGP15225 to RCh and #GGP20043 to EB); European Joint Project – Rare Diseases (EJP-RD WilsonMed grant to RSP and HZ); Italian Ministry of Health (grant # RF-2016-02362950 to RCh); AIRC, Italy (Grant #IG 17118 to RSP); Italian National Wilson Disease Organization (grant to RSP); CNR/RFBR (Russian Foundation for Basic Research) Collaboration Program, Italy and Russia (grant number 20-515-7813 to AI and EYI); Russian Science Foundation (grants # 22-24-00762 to LVP and # 20-74-10087 to EYI). We would like to acknowledge Advanced Microscopy and Imaging, Bioinformatics, High Content Screening, and Advanced Histophatology cores of TIGEM for help with data acquisition and analysis, Cathal Wilson (TIGEM) for critical reading of the manuscript, Annamaria Carissimo for bioinformatics support, Luca D'Orsi, Veronica Maffia and Edoardo Nusco (TIGEM) for technical help, Ines Mancini (University of Trento) for providing SM875.

## Author contributions

Conceived and designed the experiments: RSP and RCh; performed the experiments: RP, FC, RCr, EVP, ME, AM, GL, AG, MB, BA, AT, EC, QR, EVP and EYI; analyzed the data: RP, FC, EVP, ME, DC, AT, VN, LVP, AI, EYI, EDP, PP, HZ, QR, RCh and RP; contributed to the writing of the manuscript: RP, FC, EDP, AT, VN, EB, LVP, AI, EYI, HZ, RCh and RSP; provided technical support: EC, LS and NI.

## Competing interests

The authors disclose pending patent application priority filing (Applicant: Fondazione Telethon ETS, Inventor: RSP, Application #: IT 102024000028707) for the use of PrP as a target to treat disorders of copper metabolism and Wilson disease in particular. The specific aspect of manuscript covered in patent application includes the use of genetic and pharmacological suppression of PrP for reduction of copper toxicity.
