## [Transparent Peer Review file · Nature Communications]

Prion protein promotes copper toxicity in Wilson disease

Corresponding Author: Dr Roman S. Polishchuk

Version 1:

Reviewer comments:

Reviewer #2

(Remarks to the Author)

I commend the authors on providing a detailed response and a much improved manuscript. The study will represent a significant advancement in the field of Wilson disease and copper metabolism with a deeper understanding of the role of prion protein.

A major concern has not been resolved and in fact it might have worsened in the current revision. This reviewer appreciates how Fig. 6 has been modified to exclude mice who died because of accidental causes ("entrapment in cage elements"). However, this also reveals a poor management of mouse colonies with significant concerns for mouse wellness which cannot be ignored. In addition, it appeared that these accidental causes of mortality were not initially communicated by mouse facility staff to the principal investigator, worsening concerns about vertebrate animal colony management. The explanation about excessive dietary copper (8 ug/kg; is this kg of diet weight or kg of mouse weight?) in the diet is not convincing and if that is the case, the authors should provide information about weekly food intake. Also, they should provide the brand of the diet and explained if this was a custom-made diet.

An *Atp7b*^{-/-} mouse with an increased spontaneous mortality at 12-16 weeks of age would be transformative in the field as the related experiments could be much shorter. Instead, *Atp7b*^{-/-} mice can survive after 1 year of age unless there are concerns with colony mismanagement, infections, or dietary factors that were not properly controlled between *Atp7b*^{-/-} and *Atp7b*^{-/-} *Pmp*^{-/-} mice. The fact that the authors learnt about accidents in the colony only after the first revision, suggests that all these hypotheses are possible.

This reviewer's recommendation is to remove mortality data in Figure 6 as they are not representative of prion protein role on survival or mortality in Wilson disease.

Reviewer #3

(Remarks to the Author)

The authors have responded to my remarks.

Reviewer #4

(Remarks to the Author)

The authors have thoroughly responded to the referees' questions and suggestions and included important additional data in the manuscript. The study provides a significant mechanistic advance in understanding the connection between prion protein and ATP7B in copper trafficking.

A few last suggested corrections:

Lines 188-189: "Supporting this, we found that CTR1 levels in ATP7B-KO cells were higher than in WT cells (Supplementary Fig. 6A)." The figure shows that CTR1 levels are lower, not higher, in ATP7B-KO cells.

Lines 171 and 279: *Atp7b*^{-/-} mice are introduced as a WD model in line 171, mentioned a few times in the text after that, and then re-introduced as a WD model in lines 279-280.

Line 139 and 283: "*Atp7b*^{-/-}:*Pmp*^{-/-} hepatocyte populations" are mentioned in line 139, but the construction of the mouse

strain by crossing is only described in line 283.

Line 420: remove the comma between the subject and predicate in this line

Reviewer #5

(Remarks to the Author)

Dear Authors and Editor,

This is a robust and intriguing study on the role of prion protein in mediating copper toxicity in Wilson's disease, utilizing a comprehensive array of both in vitro and in vivo approaches. Although previous reports have explored the correlation between PrP and Wilson's Disease, to my knowledge, this is the first study demonstrating that PrP exacerbates copper toxicity in this context, making the work novel and highly relevant.

I have carefully reviewed the authors' responses to the other reviewers. The authors have done an excellent job revising the manuscript, incorporating new experiments, and thoroughly discussing the data based on their findings.

In summary, I am satisfied with the revised version of the manuscript and believe it represents a significant contribution to the understanding of prion protein's role in regulating copper homeostasis in vivo.

REFEREES' COMMENTS:

Referee #1 (Remarks to the Author):

Using a genetic screen, the authors identified the prion protein PrP as a mediator of copper uptake contributing to copper-induced toxicity in models of Wilson Disease. The article reads and well and the methodology is sound overall (see comments below). While there are some interesting observations in this study, it thoroughly rely on prior art making it challenging to fully appreciate the novelty. For example, PrP has previously been shown to promote copper uptake (ref 26-27), PrP copper-binding-sites have previously been identified (refs 24-28), the toxicity of copper to cells and in particular in WD has been reported (ref 3), various PrP splice forms have been correlated to early onset and severity of the WD phenotype implicating PrP in this pathology (refs 19-21) and copper has been shown to promote the expression of PrP (refs 22-23). The authors build on these findings to further document the causal role of PrP in WD models.

We appreciate the Reviewer's acknowledgment of the overall quality of our manuscript. However, we would like to address the concerns regarding the novelty of our findings. While it is true that several aspects of PrP's involvement with copper have been previously explored, our study presents significant new insights into the role of PrP in Wilson's disease (WD).

We would like to highlight the following novel contributions:

1. Hepatic PrP expression in Wilson disease: To our knowledge, our study is the first to demonstrate the activation of PrP expression specifically in the liver in the context of WD. Previous studies have not established this connection, making our findings a unique addition to the understanding of PrP's role in hepatic pathology associated with WD.
2. PrP's role in copper toxicity in Wilson disease: Although prior studies have suggested a role for PrP in copper homeostasis, none have directly linked PrP to the promotion of copper-induced toxicity in WD models. Our study provides the first evidence of PrP functioning in a toxic capacity rather than a protective one, as was previously hypothesized in the field. This finding challenges the long-held assumption that PrP might protect against copper-mediated oxidative damage (Brown and Besinger, 1998; Wong et al., 2000).

Furthermore, clinical studies by Merle et al. (2006) and Grubenbecher et al. (2006) on PrP variants and WD patient phenotypes produced conflicting results, which hindered further exploration of PrP's role in WD for nearly two decades. The association of the M129 variant with both delayed disease onset and more severe neurological symptoms underscored the complexity of PrP's involvement in WD but did not clarify its pathogenic role.

Our study not only establishes a direct link between PrP and copper toxicity in WD but also elucidates the underlying mechanism by which PrP exacerbates this toxicity in both cell and animal models. This mechanistic insight represents a significant advance in the field and addresses a previously unexplored area of Wilson disease pathology.

In summary, while our work builds upon existing knowledge, it also introduces critical new perspectives that have not been previously documented, thereby offering substantial novelty to the field.

Minor other points:

A recent study (PMID: 35298263) has shown that copper transport to mitochondria can promote cuproptosis. This study should be discussed to a minimum and some of the phenotypes described in their study should be investigated in the context of this submitted article.

We appreciate the Reviewer bringing attention to the recent study by Tsvetkov et al. (PMID: 35298263), which explores copper transport to mitochondria and its role in inducing cuproptosis. We agree that this pathway is relevant to our study and have now discussed cuproptosis more extensively in revised manuscript (page 13).

It is noteworthy that only one component associated with cuproptosis pathway, PDHX, emerged from our genetic screen (as mentioned on page 3). The absence of additional cuproptosis components in our findings could be attributed to the differences in the experimental conditions between our study and Tsvetkov's work. Specifically, Tsvetkov et al. utilized elesclomol, a copper ionophore, which significantly enhances copper-induced cell death by directly delivering copper to mitochondria, thereby triggering cell death at low concentrations (as low as 10 μ M). In contrast, our study did not involve the use of elesclomol; instead, copper entered the cells via membrane transporters, leading to its distribution across various cellular compartments.

Without elesclomol, copper toxicity likely arises through multiple mechanisms, not limited to cuproptosis. Our data, such as the Caspase 3/7 activation assays (Sup. Fig. 2A, B and Sup. Fig. 3B, D), suggest that apoptosis is also a significant contributor to copper-induced cell death in ATP7B-KO cells. While our study highlights these apoptotic pathways, we do not rule out the potential involvement of cuproptosis, particularly since mitochondria are heavily implicated in copper toxicity in ATP7B-deficient models (see discussion, page 13). This suggests that cuproptosis could play a role under specific conditions, and this possibility warrants further investigation.

This is especially important as the authors perform a counter screen using the pro-apoptotic drug vinblastine assuming that copper-induced toxicity is restricted to apoptosis.

We would like to clarify that our approach does not assume that copper-induced toxicity is restricted to apoptosis alone. As discussed above, we recognize that copper-mediated cell death in ATP7B-deficient cells likely involves multiple mechanisms, including but not limited to apoptosis and potentially cuproptosis (page 13).

The rationale behind using vinblastine was to introduce a toxic agent that induces cell death through a copper-independent mechanism. This strategy allowed us to differentiate between genetic screening hits that broadly inhibit cell death and those specifically involved in counteracting copper-induced toxicity. By distinguishing these effects, we aimed to identify

pathways and factors that are directly relevant to copper toxicity, rather than general pro-apoptotic mechanisms.

Other copper uptake mechanisms have recently been identified and their role in cancer demonstrated (PMID: 37100912). This study should also be discussed (in addition to ref 10) especially in light of the experiments using HepG2 cells stimulated by Oncostatin M, a known inducer of epithelial-mesenchymal transition characterized by increase of CD44 (which is expressed in HepG2 cells) and copper/iron uptake.

We welcome the Reviewer’s suggestion to discuss the CD44-mediated copper uptake pathway, as recently highlighted by Solier et al., 2023, in the context of our experiments. CD44 did not emerge as a hit in our screening for genes that reduce copper toxicity in ATP7B-KO cells (page 3). However, the Reviewer’s note regarding Oncostatin M-mediated induction of CD44 during epithelial-mesenchymal transition prompted us to investigate the expression of CD44 in our ATP7B-KO HepG2 cell line.

First, analysis of our already published RNA-seq data (Polishchuk et al., 2019) revealed that CD44 mRNA expression in ATP7B-KO HepG2 cells is several orders of magnitude lower than other copper-transporting proteins like SLC31A1 (CTR1) and PRNP (see Figure 1R below, panel A). Additionally, reverse transcription PCR confirmed that CD44 mRNA is almost undetectable in both WT and ATP7B-KO HepG2 cells, with HeLa cells showing substantial levels of CD44 transcript. Notably, treatment with Oncostatin M did not significantly upregulate CD44 expression in ATP7B-KO HepG2 cells (Figure 1R, panel B).

Figure 1R. CD44 expression in WT and ATP7B-KO cells.

A. Expression levels of CD44, PRNP and SLC31A1 (CTR1) were extracted from RNA-seq data published by Polishchuk et al. (2019). The graph shows very low expression of CD44 (mRNA counts) compared to PRNP or SLC31A1. **B.** Gel electrophoresis of cDNAs amplified from total RNA using CD44-specific primer.

Lines - M: molecular weight marker 100bp; 2-3: cDNA of HeLa cells; 4-5: cDNA of WT HepG2 cells; 6-7: cDNA of ATP7B KO HepG2 cells; 8-9: cDNA of ATP7B KO HepG2 treated with oncostatin-M; 10-11: blank samples for β -Actin and CD44 respectively. Housekeeping gene beta-Actin is expressed in all cell lines, while CD44 was amplified only in the HeLa sample. Primers from the Harvard Primer Bank:

Human CD44 fw TGCCGCTTTGCAGGTGATT

Human CD44 rev CCGATGCTCAGAGCTTTCTCC

We hypothesize that the lack of CD44 induction in our hepatic model may be due to Oncostatin M's differential role in liver cells compared to other epithelial contexts. In hepatic cells, Oncostatin M is known to promote differentiation and maturation rather than drive epithelial-mesenchymal transition, which might explain the lack of CD44 expression (Tomizawa et al., 2017; Danoy et al., 2020; van der Wouden et al., 2002).

Despite the low expression of CD44 in our model, we agree that the CD44-mediated copper uptake pathway deserves discussion. We have included additional discussion in our manuscript, noting that 'non-canonical' pathways, such as those mediated by PrP or CD44, could play significant roles in copper uptake under specific physiological or pathological conditions (Page 12).

The authors might want to comment on why DMT1 suppression does not alter normal cellular homeostasis. One would expect that cellular iron homeostasis would be affected as well, which in turn could indirectly impact on copper-related toxicity (e.g. reduced cell proliferation or reduced metabolism conferring generic protection against copper-induced toxicity).

We appreciate the Reviewer's insightful comment regarding the potential impact of DMT1 suppression on cellular homeostasis, particularly with respect to iron metabolism and its possible indirect effects on copper-related toxicity. Interestingly, our data indicate that DMT1 silencing did not significantly affect cell viability, as demonstrated by the initial points on the dose-response curve in Figure 5C.

We propose two potential explanations for this observation:

1. Residual DMT1 activity: Despite silencing, approximately 20% of DMT1 expression remains (see Figure 2R below, panel A). This residual DMT1 may still be sufficient to maintain essential iron import levels, thereby supporting cell viability even under reduced DMT1 conditions.
2. Iron overload in ATP7B-deficient cells: DMT1 suppression may be less critical for iron supply in ATP7B-deficient cells due to the elevated intracellular iron levels associated with ATP7B loss. Wilson disease is characterized by hepatic iron accumulation (Pak et al., 2021), resulting from impaired ATP7B-mediated copper delivery to ceruloplasmin. This impairment hinders ceruloplasmin's Fe(II)/Fe(III) ferroxidase activity, leading to decreased circulating iron, increased iron stores, and ultimately, iron accumulation in the liver. Consistent with these published observations, our data show nearly a two-fold increase in iron levels in ATP7B-KO cells (see Figure 2R, panel B). Thus, this iron accumulation may compensate for the reduced iron uptake due to DMT1 suppression, preventing any significant impact on cellular homeostasis or viability.

Figure 2R. DMT1 suppression and iron levels in ATP7B-KO cells.

A. qRT-PCR demonstrates reduction in DMT1 mRNA levels in ATP7B-KO line after 72h incubation with DMT1-specific siRNA (** $p < 0.001$, t-test; $n = 3$ experiments). **B.** WT and ATP7B-KO HepG2 cells were subjected to ICP-OES analysis, which shows that ATP7B knockout silencing leads to increase in intracellular iron levels in ATP7B-KO cells (** $p < 0.001$, t-test; $n = 3$ experiments).

The authors use MTT viability assay, which do not quantify cell death per se. Perhaps Annexin/PI or LDH release would be better suited.

We agree with the Reviewer that the MTT assay primarily measures cell viability rather than directly quantifying cell death. However, we would like to emphasize that we complemented the MTT assay with multiple cell death assays across all key experiments to ensure comprehensive analysis. Specifically, we employed the Live/Dead staining and the apoptotic caspase 3/7 activation assay (CellEvent assay) to directly measure cell death.

For example, during the validation of genome-wide screening hits, both the MTT and Live/Dead assays were used (Fig. 1D-F). Additionally, we assessed the impact of PRNP silencing on the resistance of ATP7B-KO cells to copper using all three methods: MTT, Live/Dead, and CellEvent assays (Fig. 2A-C; Sup. Fig. 2A, B). In the revised manuscript, we have also incorporated new data on copper-mediated cell death in primary hepatocytes, evaluated using Live/Dead and CellEvent assays (Sup. Fig. 3). Furthermore, we used CellEvent and BOBO-3 labeling to assess the role of different mutations on PrP's ability to promote copper-mediated cell death (Fig. 2B, C; Sup. Fig. 4A, B).

This comprehensive approach, combining viability and direct cell death assays, strengthens the validity of our findings.

The authors should provide a stronger rationale for the role of TFR2. This hints towards a role of iron in copper-related toxicity.

We agree with the Reviewer that the identification of TFR2 suggests a significant connection between iron metabolism and copper-related toxicity. Our findings indeed indicate that several iron-handling proteins, including STEAP1 and DMT1, may exacerbate copper toxicity in ATP7B-deficient cells, potentially by acting downstream of PrP.

In the case of TFR2, our data suggest that its role in promoting copper toxicity is independent of PrP (as shown in Supplementary Figure 7D and related results on page 9). While this points

to a distinct mechanism, the exact pathway through which TFR2 influences copper toxicity remains unclear. We believe that unraveling this mechanism will require a focused and detailed study, which we plan to pursue in future research.

One would favour ICP-MS over ICP-OES as the latter appears to quantify copper(I) only.

To the best of our knowledge this does not seem to be the case if the quantification of copper in the biological specimen is intended. ICP-OES is designed to detect and accurately quantify the total elemental content of various metals, including copper, regardless of their binding forms (e.g., whether bound to proteins) or ionic states within the sample.

In the context of copper analysis, ICP-OES does not distinguish between copper (I) and copper (II) in the biological specimen because the high-energy plasma conditions ensure that all forms of copper in this specimen are ionized uniformly. Uniform ionization means that majority of copper ions and atoms in the biological specimen are transformed to copper (I) state in the plasma. Then indeed, as Reviewer indicated, ICP-OES quantifies copper (I) as predominant chemical form in the plasma. However, it is impossible to define whether copper (I) ions in the plasma derived from copper (I) or copper (II), or even from atomic copper in the starting biological specimen. Thus, the ICP-OES analysis uses copper (I) in the high-energy plasma as a readout of the total elemental content of copper in the starting biological material.

Therefore, ICP-OES provides a reliable measure of total copper content in biological materials and does not limit its analysis to a specific oxidation state of copper.

The authors investigated the role of STEAP1 but there could be other lysosomal enzyme that can be at play here and redundant.

We appreciate the Reviewer's suggestion regarding the potential involvement of other lysosomal enzymes. In response, we investigated whether silencing STEAP2 or STEAP3, which also localize to the endo-lysosomal compartment and might function redundantly with STEAP1, could improve copper tolerance in ATP7B-KO cells. Our findings indicate that silencing STEAP2 indeed reduces copper toxicity in these cells. We have integrated these results into the revised manuscript (see Fig. 5E and the corresponding section of the results on Page 9).

P4: 'fairly safe', P5: 'discovered a while ago', P6: 'vicious cycle' read colloquial.

We have revised the text to address the Reviewer's concerns regarding the colloquial expressions. However, we would like to retain the term "vicious cycle" in the revised text, as it is commonly used in scientific literature, including in our previous publication (Chesi et al., Hepatology, 2016), unless the Editor advises otherwise.

P7: The authors comment on the role of DMT1 to export copper(I) from endo/lysosomes but this protein is a divalent metal transporter. Please clarify.

While DMT1 is indeed primarily recognized as a divalent metal transporter, existing literature indicates that it can also transport copper(I) (Arredondo et al., 2003). We have mentioned this aspect of DMT1's function in the revised manuscript on page 8.

P9: CD44 should be discussed in this context and ICP-MS would certainly provide a better readout of total copper (see comment above).

Please see our comments regarding CD44 and ICP-OES/MS above.

P9: please revise the following statement: This may be due to the activation of 'some pathways' which still allow Cu accumulation in the chronic absence of the PrP protein in Atp7b^{-/-} mice but 'the could' be rendered less toxic due to differential compartmentalization'.

This reads random, speculative and not supported by experimental data.

We understand that the statement mentioned by Reviewer may have appeared speculative and unsupported by experimental data.

In our original manuscript, we observed that acute suppression of Prnp by RNAi significantly reduced hepatic copper levels in Atp7b^{-/-} mice, whereas stable Prnp knockout did not produce the same effect, despite improved phenotype. To explain this discrepancy, we hypothesized that in the chronic absence of PrP-mediated uptake, copper has enough time to explore alternative pathways to enter Atp7B-deficient cells. These pathways could be less toxic than PrP-mediated uptake, allowing hepatic cells to adapt by modulating copper compartmentalization and sequestration.

We acknowledge the Reviewer's valid concern regarding the lack of direct experimental support for this hypothesis. Our updated data provide some evidence supporting this hypothesis (see pages 10-11 and Supplementary Fig. 9). Specifically:

1. Copper compartmentalization: Prnp knockout in Atp7b^{-/-} mice normalized mitochondrial ultrastructure and reduced electron-dense particles associated with mitochondrial copper accumulation (Zischka et al., 2011). In contrast, these particles were observed in endo-lysosomal structures in the Atp7b^{-/-} strain (Supplementary Fig. 9B), suggesting that copper may be compartmentalized differently in Atp7b^{-/-}:Prnp^{-/-} mice compared to Atp7b^{-/-} mice.
2. Copper sequestration and detoxification: Increased expression of Mt1/2 in Atp7b^{-/-}:Prnp^{-/-} mice (Supplementary Fig. 9E) may enhance copper sequestration and detoxification. Conversely, lower expression of Steap1 (Supplementary Fig. 9E) could reduce copper reduction, affecting its transport from endocytic organelles to the cytoplasm. This might result in copper remaining trapped in the endo-lysosomal system, with some recycled into systemic circulation and excreted in urine. Indeed, we found increased urinary copper in Atp7b^{-/-}:Prnp^{-/-} mice compared to Atp7b^{-/-} mice (Supplementary Fig. 9F).

These observations in *Atp7b*^{-/-}:*Prnp*^{-/-} mice resemble the effects of D-Penicillamine treatment, which promotes urinary copper excretion in Wilson's Disease patients without reducing hepatic copper levels.

We hope these clarifications and additional data address the Reviewer's concerns.

Referee #2 (Remarks to the Author):

This is a study from Petruzzelli et al. in Roman Polishchuk research group where, with an interesting series of *in vitro* and *in vivo* experiments, the authors explore a previously unexplored role of Prion protein in Wilson disease.

We would like to thank reviewer for positive evaluation of our manuscript and its scientific interest. We appreciate your recognition of the significance of our study.

This is not the first study that suggests a role of Prion protein in Wilson disease pathogenesis. Two previous papers (one from Uta Merle and one from Grubenbecher- the latter is cited in the submitted manuscript) indicated a possible, although rather weak association between Prion protein polymorphisms and the varied onset of symptoms in patients with Wilson disease.

We acknowledge the Reviewer's assessment regarding the limitations of the existing evidence on the association between PrP polymorphisms and Wilson disease. Clinical studies by Merle and Grubenbecher provide conflicting data: Merle et al. (2006) reported that homozygosity for the M129 variant of PrP is associated with a later onset of the disease, whereas Grubenbecher et al. (2006) found that homozygosity for M129 correlates with more severe neurological symptoms in Wilson disease patients. These inconsistencies, coupled with the previously believed low expression of hepatic PrP, led to a lack of further investigation into PrP-mediated mechanisms in Wilson disease.

Our study is the first to demonstrate a toxic role of PrP in cell and animal models of Wilson disease. Furthermore, we elucidate a mechanism by which PrP contributes to copper toxicity.

This is an interesting study from a prominent group of scientists well-known in the field. The *in vitro* experiments are mostly convincing, although performed only on a cell line (unfortunately, no primary cells used).

To address the concern regarding the use of a single cell line, we have now included additional data demonstrating that genetic suppression of *Prnp* reduces copper toxicity in primary hepatocytes derived from *Atp7b* knockout mice (see Supplementary Fig. 3). This addition strengthens the *in vitro* component of our study by incorporating primary cells.

The concern is about the actual *in vivo* physiological significance of prion protein. There are several important concerns about the *in vivo* studies.

1) How is this compatible with the fact that there is minimal amount of free copper in the cell? What signal would cause copper accumulation to cause prion protein to increase and further copper internalization?

We agree with the Reviewer's notion that free Cu is kept to minimal amounts inside the cell. However, a significant portion of copper exists as part of the "labile" copper pool, which is reversibly and loosely bound (Cotruvo et al., 2015; Xiao et al., 2018; Lutsenko, 2021). In the absence of ATP7B, this labile copper pool increases due to impaired copper export, both to the biosynthetic pathway and extracellular space. Using the CF4 sensor (Xiao et al., 2018), we demonstrated that this increase in labile copper occurs in ATP7B-KO cells (Fig. 2D, E). Therefore, ATP7B loss results in the accumulation of labile (easily exchangeable) copper.

How does this trigger an increase in PrP-mediated Cu uptake? Previous studies suggest that elevated intracellular copper levels are sensed by transcription factor MTF1 (reviewed in Giedroc et al., 2001), which then binds to metal-responsive elements (MREs) in the promoter of the PRNP gene, thereby enhancing its transcription (Bellingham et al., 2008). In the revised manuscript, we provide evidence that copper stimulates PRNP expression via MTF1 in ATP7B-KO cells. Specifically, silencing MTF1 prevents copper from activating PRNP transcription in these cells (Fig. 3J). This indicates that copper-dependent activation of MTF1 leads to increased expression of PrP, which subsequently enhances PrP-mediated copper uptake. While this mechanism was mentioned in the original manuscript, we have now emphasized its significance more clearly in the revised version, supported by new data (Fig. 3J).

2) The manuscript reports that a series of mutations/deletions at Cu-binding sites in prion protein were associated with reduced copper toxicity. How these induced deletions/variants correlate with previously described prion protein variants in patients with Wilson disease?

As noted by the reviewer, our manuscript reports that mutations and deletions at copper-binding sites in prion protein are associated with reduced copper toxicity. Specifically, the G54S substitution, which exacerbates Wilson disease manifestations (Forbes et al., 2014), is located near the octapeptide repeat regions of PrP. This substitution may influence the copper-binding properties of adjacent octapeptide repeat domains, potentially enhancing PrP's capacity to promote copper toxicity. Our new data support this hypothesis, showing that the homologous substitution (G53S) in mouse PrP increases copper toxicity in ATP7B-KO cells (Supplementary Fig. 4). However, a detailed structural and molecular analysis of this mutant will be necessary to further elucidate its effects.

The influence of the WD-related M129V polymorphism on PrP's copper-binding properties is less straightforward. This polymorphism occurs at a considerable distance from the copper-binding sites of PrP. According to Ott et al. (2007), the M129V polymorphism affects the distribution of PrP topological forms. The methionine at position 129 may increase the proportion of PrP molecules with copper-binding domains oriented towards the cytosol, while the valine variant reduces this orientation, resulting in more PrP molecules with copper-binding domains available for copper uptake from the extracellular space. Consequently, the V129 polymorphism could potentially enhance copper uptake and contribute to a more severe WD phenotype.

This hypothesis is consistent with clinical studies showing that the V129 variant is associated with an earlier onset of Wilson disease (Merle et al., 2006). However, it conflicts with another study suggesting that the V129 variant is linked to less severe neurological symptoms in Wilson disease patients (Grubenbecher et al., 2006).

In our experiments, the mouse homolog of the V129 variant (V128) did not significantly affect PrP's ability to promote copper toxicity in hepatic ATP7B-KO cells, although there was a trend towards increased copper toxicity in PrP V128-expressing cells (Supplementary Fig. 4). This lack of a clear impact on copper toxicity in hepatic cells suggests that the M129V polymorphism may influence the neurological phenotype of Wilson disease. Further studies in a neurological context are warranted to fully understand its implications.

3) Are there any data on actual expression of prion protein in liver biopsies from patients with Wilson disease?

We believe the Reviewer may have overlooked these data in the original manuscript (Sup Fig. 4H in old manuscript). The related part of the results says that we mined already published transcriptomics data in WD patients (Nicastro et al., 2021) and found higher mRNA levels of *PRNP* compared to healthy liver tissue (Fig. 3H).

4) How is the expression of other copper transporters in relation to prion protein variants? For example, how does ATP7A expression changes in response to prion protein KO? Also how are metallothionein levels?

We investigated how *Prnp* knockout affects the expression of copper-related genes in the liver of *Atp7b*^{-/-} mice. Our analysis revealed no significant changes in the expression levels of *Atp7a*, *Ctr1*, and *Dmt1*. However, we observed a notable reduction in *Steap1* expression, while metallothioneins 1 and 2 (*Mt1/2*) were significantly upregulated (Supplementary Fig. 9E).

The implications of these changes in the expression of copper-associated genes and their potential contribution to the observed improvement in the hepatic phenotype of *Atp7b*^{-/-}:*Prnp*^{-/-} mice are discussed in detail on pages 10-11 of the manuscript.

5) Certainly, it is very interesting that liver histology and ALT levels were improved in *Atp7b*^{-/-} and *Prnp*^{-/-} mice at 28 weeks. Remarkably, this was associated with better survival of the double KO strain compared to *Atp7b*^{-/-} mice (hybrid background). This review is concerned about the high mortality observed in *Atp7b*^{-/-} mice starting at age 6 weeks and worsening between 12-20 weeks of age. What was the cause of mortality?

This is not in agreement with the well-known and published natural history of *Atp7b*^{-/-} mice (for example, see Muchenditsi A, *Sci Rep* 2021). The authors should provide data on food intake, weight loss, and liver histology on mice, and especially cause of mortality. What was liver histology for the mice that died?

We appreciate the Reviewer's concerns and address them as follows:

1. Cause of Mortality: Evidence suggests that liver failure was the primary cause of mortality in our Atp7b^{-/-} mice. Elevated ALT levels and, occasionally, bilirubin levels were detected shortly before death, correlating with hepatitis and significant leukocyte infiltration in liver histology. Importantly, there was no significant weight loss observed in the animals prior to death.
2. Natural History of Atp7b^{-/-} Mice: We thank the Reviewer for highlighting this. We rechecked all mortality cases with our animal facility, and those few deaths that occurred before 12 weeks of age were accidental (e.g., entrapment in cage elements). Therefore, we have excluded these cases from consideration (see Fig. 6B in the revised manuscript). However, we can confirm that in our colony of Atp7b^{-/-} mice, significant mortality begins at 12-14 weeks of age.

In the past, we discussed the age and rate of mortality with different labs that have colonies of Atp7b^{-/-} mice on the same background. In some labs, the mortality rate was similar to ours, while in others, it was lower. It turns out that the main cause of such discrepancy is the quantity of copper in the diet. Our animals receive chow with 8 µg/kg of copper, whereas the dietary copper concentration was lower in labs with reduced mortality. Notably, different chow suppliers provide variable copper concentrations (always within the normal range). Indeed, using chow with a lower copper concentration (5 µg/kg) in our Atp7b^{-/-} mice resulted in decreased mortality and later disease onset. However, the animal studies presented in this manuscript were initiated with dietary copper at an 8 µg/kg concentration and were consistently conducted with the same concentration, as indicated in the revised Materials and Methods section.

On this note, Fig. 5 legend lacks several information, for example the actual number of mice has to be reported (and not just > 15), also sex of mice should be reported and both sexes should be described separately. If one sex was chosen, this choice has to be explained.

We have updated the figure legend as requested. The exact number of mice in each group is now provided. We have also separated the data by sex, and no sex-related differences in response to Prnp knockout were observed.

6) The mention of cholangiocarcinoma nodes is questionable. The nature of those nodules is not completely clarified and some could be regenerative nodules which indeed may help with mouse survival.

We agree with the Reviewer's caution regarding the interpretation of liver nodules. Our revised manuscript clarifies that while some nodules exhibit histological features consistent with normal liver parenchyma, others display characteristics indicative of cholangiocarcinoma. This includes the presence of extensive proliferating biliary ducts and strong positivity for the cholangiocarcinoma marker cytokeratin 19 (CK19). These findings align with previous reports by Lutsenko's lab (Huster et al., 2006). We have included CK19 immunohistochemistry data (Supplementary Fig. 8) in the revised manuscript to support our identification of cholangiocarcinoma-like nodules.

7) The major concern is the lack of change/improvement in hepatic copper concentration in Atp7b^{-/-} mice after Prnp^{-/-} knockout. The authors do not show any data on hepatic copper staining and distribution, copper in the urine and feces, no TEM images on in vivo mitochondria morphology to support their hypothesis.

We agree with the Reviewer that our initial manuscript lacked experimental support for our hypothesis regarding lack of change/improvement in hepatic copper concentration after Prnp knockout in Atp7b^{-/-} mice. Although acute Prnp suppression via specific siRNA significantly reduced hepatic copper levels in Atp7b^{-/-} mice (Supplementary Fig. 9D), a stable Prnp knockout did not produce a similar decrease (Supplementary Fig. 9C), despite improvements in phenotype (Fig. 6). We hypothesize that in the chronic absence of PrP-mediated copper uptake, copper could have sufficient time to explore alternative pathways for entry into Atp7b-deficient cells. These pathways could be less toxic (compared to PrP-mediated uptake), allowing hepatic cells to adapt by modulating copper compartmentalization and sequestration.

In the revised manuscript, we made every effort to collect additional in vivo experimental data, which provide support to our hypothesis. We hope that Reviewer appreciates that collecting these data was particularly challenging due to the extremely stringent animal work regulations in Italy.

First, we observed that Prnp knockout in Atp7b-deficient mice led to normalization of mitochondrial ultrastructure and the absence of electron-dense particles, which correlate with mitochondrial copper accumulation (Zischka et al., 2011). In contrast, such particles were observed in endo-lysosomal structures (Supplementary Fig. 9A, B). These observations suggest that copper might be compartmentalized differently in Atp7b^{-/-}:Prnp^{-/-} animals compared to the Atp7b^{-/-} strain.

Second, we found that Prnp knockout in Atp7b^{-/-} mice led to increased expression of metallothioneins 1 and 2 (Supplementary Fig. 9E), which should enhance copper sequestration and detoxification. Finally, we found that Prnp knockout in Atp7b^{-/-} mice resulted in the lower expression of Steap1 (Supplementary Fig. 9E). Decrease in Steap1 expression may not favor copper reduction for subsequent transmembrane transport from endocytic organelles to the cytoplasm. Therefore, Cu remaining in the endocytic system could explain why we still observe high Cu levels in the liver of Atp7b^{-/-}:Prnp^{-/-} animals. We think, however, that at least part of this remaining Cu could be recycled back into systemic circulation and neutralized through urinary excretion. Indeed, we found higher urinary copper levels in Atp7b^{-/-}:Prnp^{-/-} animals compared to Atp7b^{-/-} mice (Supplementary Fig. 9F). Notably, this effect resembles that of D-penicillamine, which promotes urinary copper excretion in Wilson's Disease patients without reducing hepatic copper levels. As the Reviewer suggested, we also checked copper levels in feces but did not find any significant differences, indicating that biliary excretion is not involved.

We hope the Reviewer finds that these new data provide a partial explanation for the observed improvement in phenotype in double knockout mice, despite unchanged hepatic copper levels.

Referee #3 (Remarks to the Author):

The manuscript shows evidence for the role of the prion protein in models of Wilson disease. The manuscript is well presented and experiments are described in detail. I would like to ask the authors to respond to the following critiques. In the initial screening the use of ATP7B-KO HepG2 cells is described. This is a human cell line and the expression of prions should be checked. We always assume that we need to induce disease in cells in order to harbor prions but it would be beneficial to actually show their absence. In addition, the mutational analysis of the octarepeat region and the non-octarepeat copper binding site is carried out with a mouse prion protein molecule. It would have been more coherent to use a human prion protein and again I would like to suggest the analysis of putative prion presence. It has been shown that mutation in these regions may harbor spontaneous or higher presence of prions.

We would like to thank Reviewer for positive evaluation of our manuscript.

We chose to express mouse PrP molecules in human ATP7B-KO HepG2 cells for biosafety reasons, to minimize the risk that the mutants could spontaneously convert into prions capable of infecting humans.

At the Reviewer's request, we have now checked for the presence of detergent-insoluble and proteinase K (PK)-resistant PrP, which are biochemical hallmarks of the infectious PrP^{Sc} isoform, in wild-type and mutant PrP-transfected ATP7B-KO HepG2 cells. The results, presented in Figure R3 below, show no evidence of insoluble or PK-resistant PrP in HepG2 cells. In contrast, these features are readily detectable in mouse neuroblastoma N2a cells chronically infected with the 22L prion strain, which we analyzed in parallel as a positive control.

Figure R3. Wild-type and mutant PrPs do not acquire the biochemical properties of PrP^{Sc} in ATP7B-KO HepG2 cells. ATP7B-KO HepG2 cells expressing wild-type PrP or mutants with a deletion of the octapeptide repeat region (Δ OR), or substitutions of alanines for histidines in the OR (Ala1) or both OR and non-OR regions (Ala2), were lysed in the presence of non-denaturing detergents (0.5% Triton X-100, 0.5% Na deoxycholate in 5 mM Tris-HCl pH 7.4, 150 mM NaCl, 5 mM EDTA). Detergent insolubility (A) and PK resistance (B) of PrP were assayed as described (Masone et al., 2023). ScN2a cells chronically infected with 22L prions served as a positive control. S, supernatant; P, pellet.

These findings indicate that the mutant PrP molecules used in our study do not spontaneously acquire the biochemical properties of PrP^{Sc} in ATP7B-KO HepG2 cells. While a bioassay in mice would be necessary to definitively exclude the presence of prions in these cells, we would like to note that, to the best of our knowledge, *bona fide* prion infectivity has been reported to spontaneously arise only in RK13 rabbit kidney cells expressing a deletion of the highly conserved ¹⁹⁰HTVTTTT¹⁹⁶ segment of ovine PrP (PMID: 32788216, DOI: 10.1074/jbc.RA120.014738). RK13 cells are one of the few immortalized cell lines known to support prion replication (PMID: 31943194, DOI: 10.1111/jnc.14956), and it is currently unknown whether HepG2 cells have this capability.

Lastly, whether ASO strategy may be beneficial to prion diseases is still to fully be proven, let alone the use of this approach to the treatment of Wilson disease.

Our data suggest that PrP plays a role in mediating copper toxicity, indicating that PrP-lowering could be a potential therapeutic strategy for Wilson disease. However, we agree with the reviewer that, without proof of efficacy for the anti-PrP ASO, it is premature to specifically reference this approach. Therefore, we have removed the statement regarding the Ionis clinical trial and instead cited a recent review of the main experimental PrP-targeting strategies currently being pursued.

POINT BY POINT REPLY TO REVIEWERS

Reviewer #2 Comments:

I commend the authors on providing a detailed response and a much improved manuscript. The study will represent a significant advancement in the field of Wilson disease and copper metabolism with a deeper understanding of the role of prion protein.

We would like to thank the Reviewer for overall positive evaluation of our effort to improve the manuscript during revision.

A major concern has not been resolved and in fact it might have worsened in the current revision. This reviewer appreciates how Fig. 6 has been modified to exclude mice who died because of accidental causes (“entrapment in cage elements”). However, this also reveals a poor management of mouse colonies with significant concerns for mouse wellness which cannot be ignored. In addition, it appeared that these accidental causes of mortality were not initially communicated by mouse facility staff to the principal investigator, worsening concerns about vertebrate animal colony management.

An *Atp7b*^{-/-} mouse with an increased spontaneous mortality at 12-16 weeks of age would be transformative in the field as the related experiments could be much shorter. Instead, *Atp7b*^{-/-} mice can survive after 1 year of age unless there are concerns with colony mismanagement, infections, or dietary factors that were not properly controlled between *Atp7b*^{-/-} and *Atp7b*^{-/-} *Pmp*^{-/-} mice.

The explanation about excessive dietary copper (8 ug/kg; is this kg of diet weight or kg of mouse weight?) in the diet is not convincing and if that is the case, the authors should provide information about weekly food intake. Also, they should provide the brand of the diet and explained if this was a custom-made diet.

We fully understand the Reviewer's concern regarding mortality of *Atp7b*^{-/-} mice. However, we would like to emphasize that our animal house operates in accordance with specific-pathogen-free (SPF) standards. This strategy enhances the efficiency of the facility and minimizes the risk of pathogen-related interference, ensuring more reliable experiments and outcomes. Therefore, we can confidently rule out infections as a cause of the animal deaths.

While our animal house staff strives to provide the best possible care, the accidental death of some animals is inevitable, although it occurs with very low incidence and can even affect wild-type animals. We agree with the reviewer that these few cases of accidental mortality should not have been included in the data, and we recognize this oversight. However, this error does not imply that the animals were poorly cared for or mishandled. Given this, we cannot agree with the reviewer's suggestion that the increased spontaneous mortality of *Atp7b*^{-/-} mice at 12-16 weeks of age is due to inadequate care or infections.

One possible factor could be dietary copper. As mentioned in our previous response, when we used a diet with lower copper content, spontaneous mortality occurred at a much later

age (around 24 weeks). Our diet, purchased from ssniff Spezialdiäten GmbH, contains 8 mg of copper per 1 kg, according to the manufacturer. This is a standard diet widely used in various facilities globally. Another factor that may vary across animal care facilities is the copper content in the water supply.

We believe the variation in the age of death of *Atp7b*^{-/-} mice across different colonies mirrors what can be observed in patients. It is well-documented that patients with the same *Atp7b* mutation(s)—even monozygotic twins—can present with significantly different phenotypes, often influenced by environmental factors such as diet and lifestyle. Similarly, differences in the onset of mortality between colonies may be driven by environmental factors, including diet and drinking water. Investigation of such environmental factors would be an interesting next step in Wilson disease research.

Finally, we want to emphasize that both single (*Atp7b*^{-/-}) and double (*Atp7b*^{-/-}: *Prnp*^{-/-}) animals were housed under identical conditions in our facility—same room, diet, water, and care. Therefore, the improvement in phenotype can be attributed solely to *Prnp* suppression, not differences in animal management.

Reviewer #4 (Remarks to the Author):

The authors have thoroughly responded to the referees' questions and suggestions and included important additional data in the manuscript. The study provides a significant mechanistic advance in understanding the connection between prion protein and ATP7B in copper trafficking.

We sincerely thank the Reviewer for his/her positive comments and the recognition of our efforts to address the referees' questions and suggestions.

A few last suggested corrections:

Lines 188-189: "Supporting this, we found that CTR1 levels in ATP7B-KO cells were higher than in WT cells (Supplementary Fig. 6A)." The figure shows that CTR1 levels are lower, not higher, in ATP7B-KO cells.

Lines 171 and 279: *Atp7b*^{-/-} mice are introduced as a WD model in line 171, mentioned a few times in the text after that, and then re-introduced as a WD model in lines 279-280.

Line 139 and 283: "*Atp7b*^{-/-}:*Prnp*^{-/-} hepatocyte populations" are mentioned in line 139, but the construction of the mouse strain by crossing is only described in line 283.

Line 420: remove the comma between the subject and predicate in this line.

The text of the manuscript was corrected according to the Reviewer's remarks.